# Two-photon imaging of neuronal activity in motor cortex of marmosets during upper-limb movement tasks

Teppei Ebina [1,2], Yoshito Masamizu[1,2,3], Yasuhiro R. Tanaka[1,2], Akiya Watakabe[4], Reiko Hirakawa[2,5], Yuka Hirayama[1], Riichiro Hira [2,3], Shin-Ichiro Terada[1,2], Daisuke Koketsu[6], Kazuo Hikosaka[7], Hiroaki Mizukami[8], Atsushi Nambu[3,6], Erika Sasaki[5,9], Tetsuo Yamamori[4] & Masanori Matsuzaki [1,2,3,10]

Two-photon imaging in behaving animals has revealed neuronal activities related to behavioral and cognitive function at single-cell resolution. However, marmosets have posed a challenge due to limited success in training on motor tasks. Here we report the development of protocols to train head-fixed common marmosets to perform upper-limb movement tasks and simultaneously perform two-photon imaging. After 2–5 months of training sessions, head-fixed marmosets can control a manipulandum to move a cursor to a target on a screen. We conduct two-photon calcium imaging of layer 2/3 neurons in the motor cortex during this motor task performance, and detect task-relevant activity from multiple neurons at cellular and subcellular resolutions. In a two-target reaching task, some neurons show direction-selective activity over the training days. In a short-term force-field adaptation task, some neurons change their activity when the force field is on. Two-photon calcium imaging in behaving marmosets may become a fundamental technique for determining the spatial organization of the cortical dynamics underlying action and cognition.

[1] Department of Physiology, Graduate School of Medicine, The University of Tokyo, Tokyo 113-0033, Japan. [2] Division of Brain Circuits, National Institute for Basic Biology, Aichi 444-8585, Japan. [3] School of Life Science, SOKENDAI (The Graduate University for Advanced Studies), Aichi 444-8585, Japan. [4] Laboratory for Molecular Analysis of Higher Brain Function, RIKEN Center for Brain Science, Saitama 351-0198, Japan. [5] Central Institute for Experimental Animals, Kanagawa 210-0821, Japan. [6] Division of System Neurophysiology, National Institute for Physiological Sciences, Aichi 444-8585, Japan. [7] Department of Sensory Science, Faculty of Health Science and Technology, Kawasaki University of Medical Welfare, Okayama 701-0193, Japan. [8] Division of Genetic Therapeutics, Center for Molecular Medicine, Jichi Medical University, Tochigi 329-0498, Japan. [9] Advanced Research Center, Keio University, Tokyo 160-8582, Japan. [10] International Research Center for Neurointelligence (WPI-IRCN), The University Tokyo Institutes for Advanced Study, Tokyo 113-0033, Japan. These authors contributed equally: Teppei Ebina, Yoshito Masamizu. Correspondence and requests for materials should be addressed to M.M. (email: mzakim@m.u-tokyo.ac.jp)

n vivo two-photon functional imaging has revealed the cellular mechanisms underlying a variety of brain functions in rodents and other small animals. When such imaging is combined with a genetically encoded calcium indicator (GECI), it allows the detection of long-term plasticity and stability in the activity of individual neurons during motor learning and sensory experience in rodents[1–5], and can do so in a layer- and cell type-specific manner. Thus, two-photon calcium imaging has the potential to be a powerful tool for revealing the principles of the long- and short-term spatiotemporal organization of the neuronal networks underlying action, cognition, and perception[6]. However, cortical organization and functioning in primates is more complex than in rodents. Non-human primate research is therefore indispensable in gaining an understanding of neuronal computation in the human brain. Recently, several groups have applied two-photon calcium imaging to awake non-human primates[7–10]. However, no study has succeeded in two-photon imaging of non-human primates participating in behavioral tasks involving upper-limb movement, such as reaching tasks. When accompanied by electrical recording of multiple neurons, these tasks are used to investigate the neuronal computation and dynamics that underlie motor control[11–14]. A full understanding of these neuronal dynamics of real cortical circuits requires further data on the spatial organization of neuronal activity: layer-specific and cell type-specific activity and axon-dendrite relationships[6]. Therefore, the establishment of two-photon calcium imaging in behaving non-human primates would be highly beneficial.

In terms of imaging, the common marmoset, a New-World monkey with the essential features of primate cortical organization[15–19], has an advantage over other monkeys in that it has a relatively small and flat (lissencephalic) cerebral cortex with a thickness of approximately 1.5–2.0 mm. This suggests that multiple cortical areas and layers of the marmoset brain may be accessible to imaging methods developed in mice[20]. Therefore, we previously established a two-photon calcium imaging technique for the neocortex of anesthetized head-fixed marmosets, and succeeded in detecting neuronal responses to upper-limb stimuli at cellular and subcellular resolutions[21]. Furthermore, although it is less dexterous than the macaque, the marmoset possesses a large behavioral repertoire[20]. A drug-administered marmoset model of Parkinson's disease exists[22,23], and the generation of transgenic marmosets expressing GECIs in the brain was recently reported[24]. Thus, an obvious next challenge is to combine two-photon calcium imaging of awake head-fixed marmosets, with the behavioral tasks used in other primates. However, the only behavioral tasks that have previously been reported for the head-fixed marmoset are saccade and licking tasks[25,26], and the teaching of upper-limb movement tasks to head-fixed marmosets is considered to be difficult[26]. To address this issue, we developed a novel behavioral apparatus, which restrains the marmoset in a chair, and trained the animal to control a two-dimensional (2D) manipulandum to move a cursor on a monitor. We demonstrate that head-fixed marmosets can learn internally triggered and external stimulus-triggered reaching tasks, and that neuronal

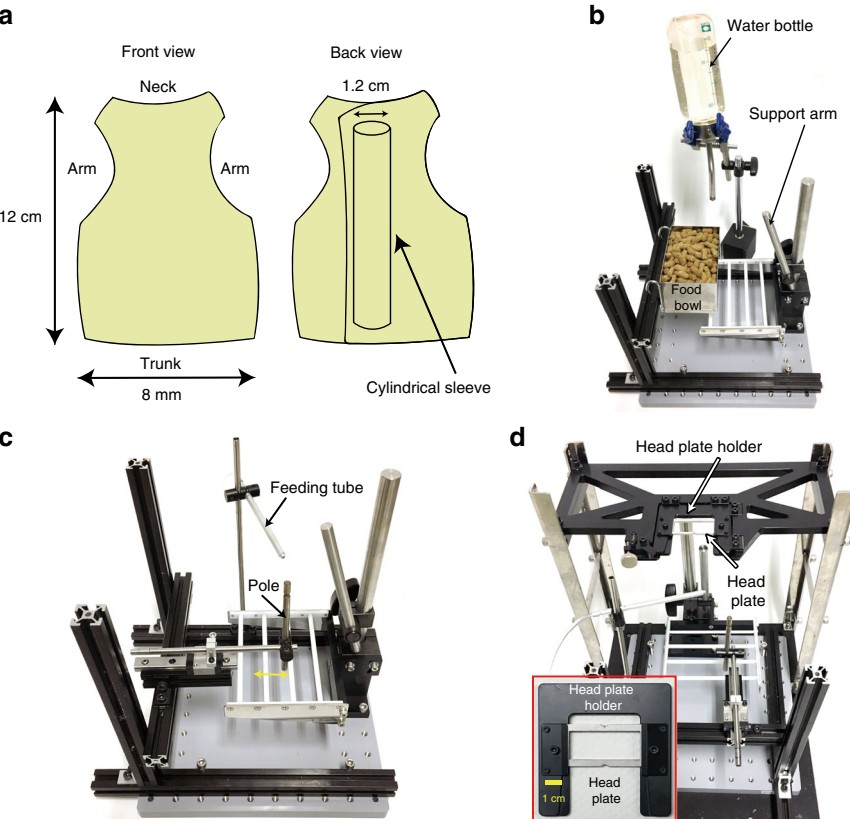

**Fig. 1** Design of the task apparatus. **a** Layout of the jacket modified from Schultz-Darken et al.[27]. The jacket has openings for the neck, arms, and trunk to pass through. The girth of the trunk was adjusted with a hook-and-loop fastener. The illustrations of the back view show the cylindrical sleeve used to restrain the trunk. **b** Apparatus for habituation to trunk constraint. The support arm was inserted through the cylindrical sleeve and restrained the trunk of the marmoset. The support arm and the cylindrical sleeve were fixed by a clip. The marmoset grasped the white scaffolding with leg paws, ate food pellets from the food bowl with upper limbs, and drank drops from the water bottle. **c** Apparatus for the self-initiated pole-pull task. The yellow double-headed arrow indicates the range in which the pole could move (3.5 cm). **d** A head plate was clamped by a head plate holder (inset), and the head plate holder was clamped by the apparatus

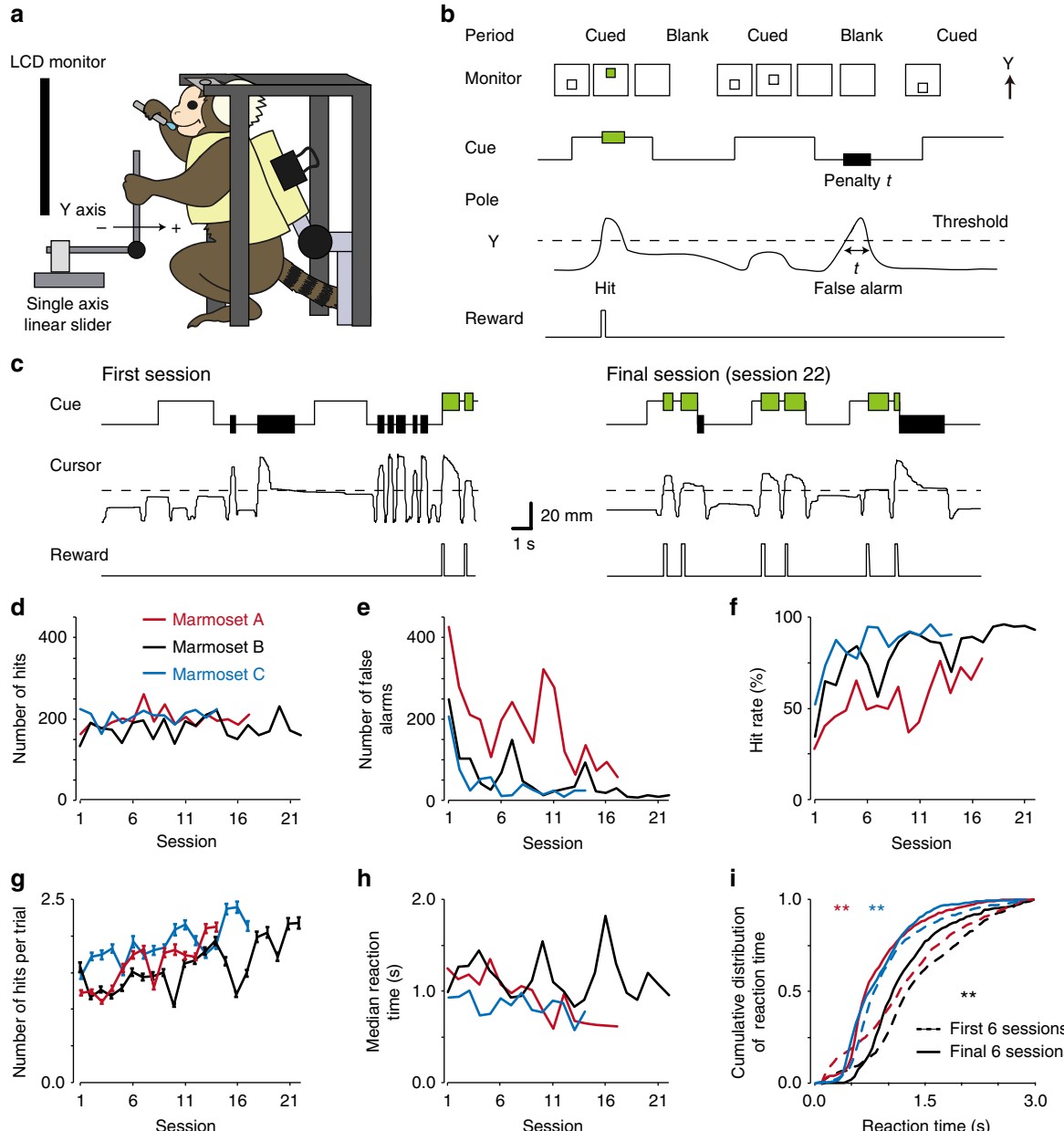

**Fig. 2** Learning of visually cued pole-pull task. **a** Scheme of the task apparatus and head-fixed marmoset. **b** Schematic diagram of the task. **c** Example of pole trajectories from marmoset B. **d**–**g** Time course of the hit number (**d**), the false alarm number (**e**), the hit rate (**f**), and the hit number per trial (**g**) across sessions. In **g**, only trials with at least one hit event were analyzed ($n = 73$–183 trials for each session). Error bars indicate SEM. Spearman correlation coefficients (CCs) between the hit number and training session were 0.35, 0.09, and 0.07 (0.22, −0.04, and 0.33 without the initial session) for marmosets A, B, and C, respectively, $P > 0.05$ for all cases. CCs between the false alarm number and training session were −0.72, −0.79, and −0.58 for marmosets A, B, and C, respectively, $P < 0.01$ for marmosets A and B, $P < 0.05$ for marmoset C. Without the initial session, CCs were –0.66, –0.76, and –0.47 for marmosets A, B, and C, respectively, $P < 0.01$ for marmosets A and B, $P = 0.10$ for marmoset C. CCs between the hit rate and training session were 0.72, 0.84, and 0.67 (0.66, 0.82, and 0.59 without the initial session) for marmosets A, B, and C, respectively, $P < 0.01$ for marmosets A and B, $P < 0.05$ for marmoset C. CCs between the hit number per trial and training session were 0.81, 0.71, and 0.81 (0.77, 0.77, and 0.77 without the initial session) for marmosets A, B, and C, respectively; $P < 0.01$. **h** Median reaction time. **i** The cumulative distribution of reaction time. For marmoset A: $1203 \pm 29$ ms in the first six sessions vs. $833 \pm 21$ ms in the final six sessions, $n = 664$ and 578 trials, respectively, $**P < 0.01$, Wilcoxon rank-sum test. For marmoset B: $1396 \pm 26$ ms vs. $1207 \pm 25$ ms, $n = 763$ and 569 trials, respectively, $**P < 0.01$. For marmoset C: $1032 \pm 18$ ms vs. $904 \pm 17$ ms, $n = 952$ and 673 trials, respectively, $**P < 0.01$

activities relevant to the upper-limb movements can be detected by two-photon imaging over periods of minutes and days.

## Results

**Training of upper-limb movement tasks without head fixation.** Physical constraint of the neck and waist may make it difficult for head-fixed marmosets to learn an upper-limb movement task[26].

We therefore used a soft jacket[27] to restrain the trunk of the animal (Fig. 1a). In this study, we trained four adult marmosets, step by step, to perform upper-limb movement tasks with head fixation (Supplementary Fig. 1). First, we habituated the marmosets to an experimental apparatus with jacket restraint for 10–60 min (one session) per day. Food and water were restrained during the habituation and training days. In the initial training

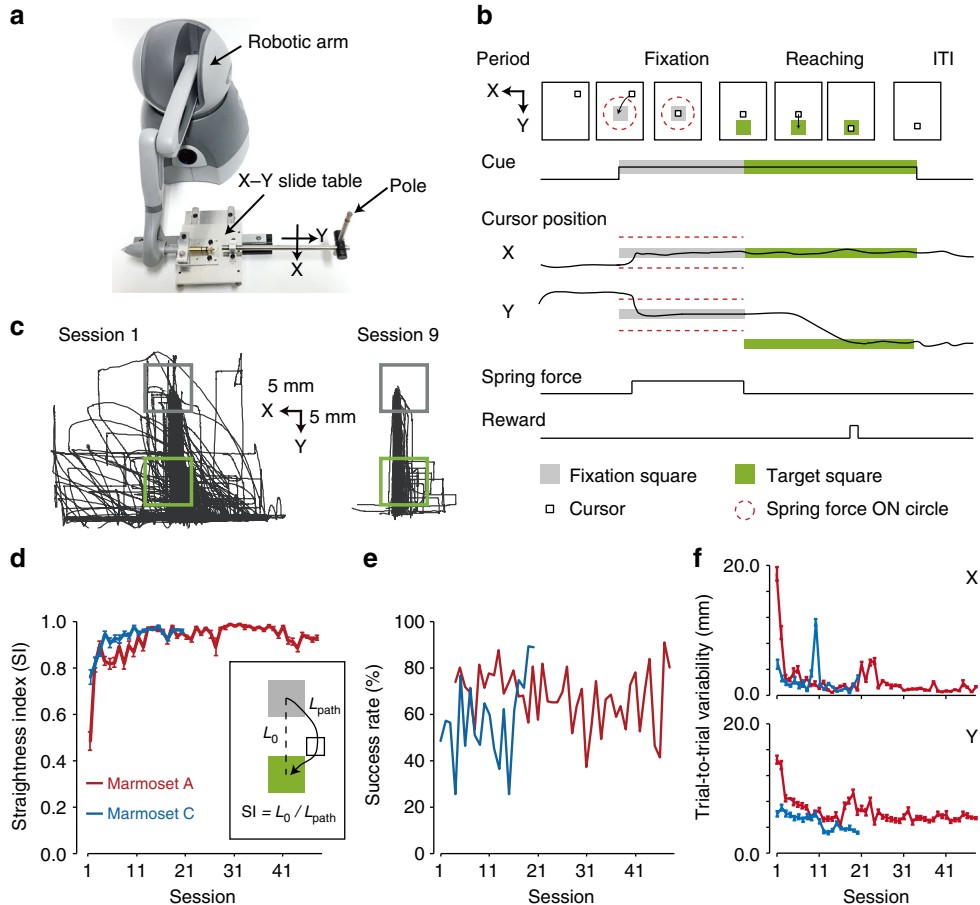

**Fig. 3** Learning of the one-target reaching task. **a** An X–Y slide table to enable marmosets to control the pole on a 2D working space (53 mm for the X-axis and 90 mm for the Y-axis). A robotic arm was connected to the table. **b** The task consisted of fixation and reaching periods, and an inter-trial interval (ITI). During the reaching period, a target (green) was presented and marmosets used the manipulandum to move the cursor from a fixation square (gray) to the target and hold it for 10–300 ms to obtain a reward. **c** Reaching trajectories in sessions 1 and 9 in marmoset A. Each trajectory for each trial is overlaid. Gray and green boxes indicate the fixation and target squares, respectively. **d** Time course of the mean straightness index (SI) of the successful reaching trajectory in marmosets A (red) and C (blue). CCs between SI and the training session were 0.48 (0.45 without the initial session) for marmoset A ($P < 0.01$) and 0.70 (0.65 without the initial session) for marmoset C ($P < 0.01$). **e** Time course of the success rate. The success rate was calculated by dividing the number of rewarded trials by that of all trials. The success rate in marmoset C was relatively low until session 15, because marmoset C had difficulties in stopping and holding the cursor within the target square, and the cursor frequently passed through it, even though the trajectory became straighter. **f** Time course of the trial-to-trial variability (see Methods) of X (top) and Y (bottom) coordinates for the successful reaching trajectories in marmosets A (red) and C (blue). The CCs between mean of the root mean square deviations (RMSDs) of X and Y coordinates and the session number in marmoset A were $-0.58$ ($-0.55$ without the initial session) and $-0.55$ ($-0.52$ without the initial session), respectively, $P < 0.01$, while for marmoset C they were $-0.44$, $P = 0.054$ ($-0.36$ without the initial session, $P = 0.12$), for the X coordinate, and $-0.85$ ($-0.83$ without the initial session), $P < 0.01$, for the Y coordinate

sessions, marmosets were allowed to freely access food in a bowl and water from a bottle placed in front of them (Fig. 1b), and within six sessions they became accustomed to the apparatus and came to quietly eat and drink. We then attached a small metal dish containing a food pellet to a linear slide, and put it in front of the marmoset. To retrieve the pellet from the dish, the animal had to reach out its left arm to pull the edge of the dish along the slider (dish-pull task). After 1–4 sessions of training on the dish-pull task, even when a pellet was not placed on the dish, the marmosets pulled the dish to request a pellet. We then trained the marmosets to perform a self-initiated pole-pull task (Fig. 1c), in which they had to pull a pole attached to the slider beyond a threshold position to obtain a drop of apple juice, which was delivered from a fixed feeding tube. Within 3–17 sessions, the marmosets pulled the pole and licked the drop from the tube without rolling their head to look for the pole or the tube (Supplementary Fig. 1).

**Self-initiated pole-pull task in head-fixed marmosets.** Next, we trained the animals to perform the self-initiated pole-pull task with their heads fixed (Fig. 1d and Supplementary Fig. 2). Even with the head fixation, the marmosets were able to quietly obtain similar numbers of rewards as they did before the fixation (Supplementary Fig. 3). In the final session, the number of reward deliveries was $90.0 \pm 35.3$ ($n = 4$ marmosets) and the training duration was $22.8 \pm 12.6$ min ($n = 4$ marmosets). These results demonstrate that head-fixed jacket-mediated trunk-restrained marmosets can perform a self-initiated pole-pull task within 1–2 months of the start of habituation.

**Visually cued pole-pull task.** Next, we trained three of these four marmosets to become skilled in an externally triggered cued pole-pull task performed under head fixation. In this task, to obtain a reward the animals had to use their left upper limb to pull the

pole beyond a threshold position while a visual cue was presented on a LCD monitor in front of the animal's head (Fig. 2a, b; cued period). During the cued period, a white cursor on the monitor moved upward, depending on the position of the handling pole with the color of the cursor turning to green when the pole position exceeded a threshold. A blank period, in which the cue was not presented, was inserted between cued periods. When the animal pulled the pole beyond the threshold during a blank period, the duration of that blank period was extended as a penalty (Fig. 2b, c, see Methods for details). The total number of beyond-threshold pole-pull events during the cued period (hits) did not change across the duration of the training (approximately 200 events in each training session; Fig. 2d), whereas the total number of pull events beyond the threshold during the blank periods (false alarms) decreased (Fig. 2e, mean ± SEM, 293.7 ± 67.3 at the initial session and 32.0 ± 14.8 at the final session, $n = 3$ marmosets). This increased the hit rate, calculated as (hits/[hits + false alarms]) × 100%, from 38% ± 7% to 87% ± 5% (mean ± SEM, $n = 3$ marmosets, Fig. 2f). In addition, as the number of rewards was not limited during a cued period, the animals increased their number of hits per cue period throughout training (Fig. 2g). From the first six sessions to the final six sessions, the reaction time from cue onset to the first pole pull shortened from 1054 ± 95 ms to 782 ± 86 ms (mean ± SEM, $n = 3$ marmosets, Fig. 2h, i). These results demonstrate that the head-fixed marmosets could wait without upper-limb movement until the visual cue appeared, and were able to learn the visual cue-triggered movement task within 1 month of the start of training.

**One-target reaching task.** The most commonly used task for studying the motor control system in primates is a hand/arm reaching task, for example, moving the arm and touching the fingertip to a target, or controlling a robotic arm to move a cursor to a target on a monitor. Therefore, we tested whether the marmosets could learn to use a 2D manipulandum to move a cursor to a target (one-target reaching task). A green target square was presented straight below a fixation square. The bottom of a pole was linked to a robotic arm on an X–Y slide table (Fig. 3a and Supplementary Fig. 4). Pulling the pole towards the animal moved the cursor in the Y-axis direction on the monitor (that is, towards the target), while a leftward movement of the pole moved the cursor in the X-axis direction on the monitor. Marmosets A, C, and D were trained to control the manipulandum with their left upper limb, to move the cursor from the fixation square to the target square and hold it inside the target to obtain a drop of juice (Fig. 3b). After the reward was given (successful trial) or the cursor moved outside the monitor or the target (failed trial), the pole was returned to the fixation point by a mechanical force while the cursor was simultaneously moved to the fixation square, before the next trial was started. As marmoset D performed this task for only five sessions, we did not analyze the behavioral changes in marmoset D (Supplementary Fig. 1; see Methods for details). Although the cursor trajectory was not straight during session 1 (Fig. 3c, left; Supplementary Movie 1), it became closer to a straight line through training (Fig. 3c, right; Supplementary Movie 2). Improvement in performance was estimated as change in the length between the start and end points of the cursor divided by the length of its actual trajectory (straightness index; SI = 0.48 and 0.76 for marmosets A and C, respectively, at session 1, and 0.90 and 0.94 for marmosets A and C at session 9). The SI exceeded 0.9 after 3–4 training sessions (Fig. 3d). In consecutive sessions 47 and 48 for marmoset A and consecutive sessions 19 and 20 for marmoset C, the success rate was sustained at >80% (Fig. 3e). The trial-to-trial variability in successful reaching of the trajectory decreased as the training progressed (Fig. 3f). Thus, it

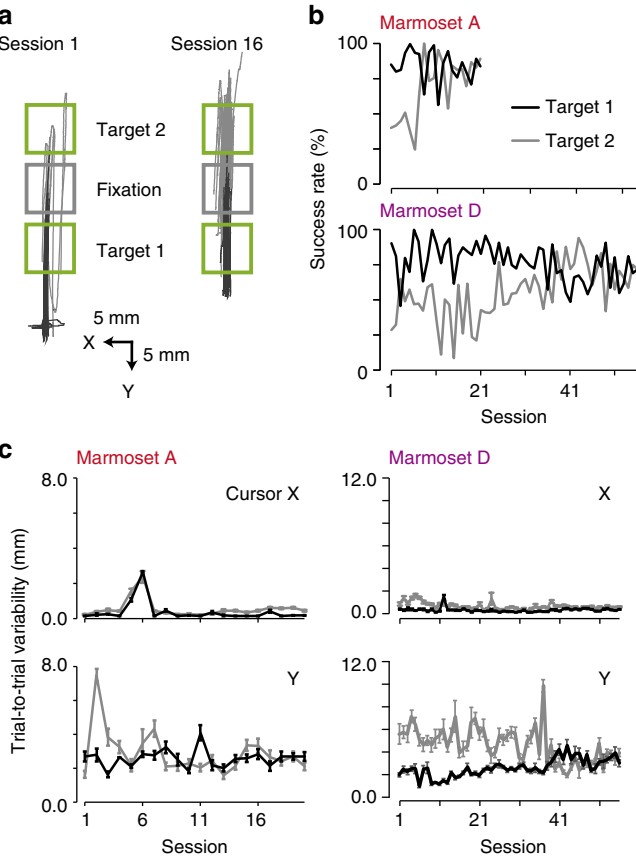

**Fig. 4** Learning of the two-target reaching task. **a** Reaching trajectories in sessions 1 and 16 of the two-target reaching task in marmoset A. Black and gray solid lines represent reaching trajectories for the targets below (target 1) and above (target 2) the fixation square, respectively. Other conventions are the same as in Fig. 3c. **b** Time course of the success rates for different targets in marmosets A (top) and D (bottom). CCs between the success rate for target 1 and session number were −0.23 for marmoset A, $P = 0.32$, and −0.44 for marmoset D, $P < 0.01$. CCs between the success rate for target 2 and session number were 0.68 for marmoset A, $P < 0.01$, and 0.61 for marmoset D, $P < 0.01$. **c** Time course of the trial-to-trial variability of the successful reaching trajectory to targets 1 (black) and 2 (gray) in marmosets A (left) and D (right). For marmoset A, CCs between the variability of the reaching target 1 and session number were −0.05 and 0.01, $P = 0.84$ and $P = 0.97$, for X and Y coordinates, respectively. CCs for the variability of reaching target 2 were 0.25 and −0.18, $P = 0.27$ and $P = 0.44$, for X and Y coordinates, respectively. For marmoset D, CCs for the variability of reaching target 1 were 0.08 and 0.72, $P = 0.54$ and $P < 0.01$, for X and Y coordinates, respectively. CCs for the variability of reaching target 2 were −0.39 and −0.64, $P < 0.01$ and $P < 0.01$, for X and Y coordinates, respectively. Marmoset D demonstrated a slow learning rate for reaching target 2, with the variability for reaching target 1 increasing. This might be because marmoset D had performed the one-reaching task with a force field for approximately 30 days, and had become heavily habituated to reaching target 1

was demonstrated that the head-fixed marmoset can learn to move a cursor to a target using a 2D manipulandum.

**Two-target reaching task.** Marmosets A and D were trained to perform a two-target reaching task (Supplementary Fig. 1) in which a second green target (target 2) was introduced above the fixation square, and one of the two targets was presented in each trial (Fig. 4a). In session 1, the success rate was high (81.3% for marmoset A and 90.4% for marmoset D) for the target in the Y-

axis direction (target 1), because the marmosets had already learned how to reach it (one-target reaching task in marmoset A and adaptation task in marmoset D, described below; Fig. 4b). By contrast, the success rate was low for target 2 (Fig. 4b). As the training progressed, the success rates for movements to target 2 significantly increased in both marmosets (Fig. 4b), although the success rate for reaching target 1 decreased in marmoset D (Fig. 4b). During the sessions, the trial-to-trial variability for successfully reaching the trajectory varied within low values (<2 mm for X coordinates and <6 mm for Y coordinates) in both marmosets (Fig. 4c). This may be because the marmosets had already learned the straightforward movement to target 1. During the last five sessions, the variability for either coordinate did not differ between the two marmosets (target 1: X coordinate, 0.20 ± 0.05 mm in marmoset A and 0.31 ± 0.04 mm in marmoset D, $n = 5$ sessions, $P = 0.09$; Y coordinate, 2.80 ± 0.12 mm and 3.13 ± 0.17 mm, $P = 0.09$; target 2: X coordinate, 0.53 ± 0.04 mm and 0.56 ± 0.04 mm, $n = 5$ sessions, $P = 0.84$; Y coordinate, 2.83 ± 0.19 mm and 3.62 ± 0.40 mm, $P = 0.15$). These results demonstrate that a head-fixed marmoset can learn to skillfully control a 2D manipulandum to move a cursor to two targets.

**Reaching task with force-field perturbation**. Finally, we introduced a force field[11,13,28] to the one-target reaching task (force-field adaptation task; Fig. 5a, b). We further trained the three marmosets A, C, and D (Supplementary Fig. 1; see Methods for details). To ensure efficient learning of the association between the manipulandum position and the reward delivery[29], we introduced an L-shape spout pole, the bottom of which was attached to the X–Y slide table, and the top of which delivered a drop of juice (Fig. 5a). When the marmosets pulled the pole to move the cursor to the target, the tip of the pole was pulled closer to the mouth, and when the cursor entered the target rectangle, a reward was delivered from the tip. A velocity- (in the Y-axis direction) dependent force field was set orthogonal to the direction from the fixation center to the target center (the X-axis direction), and the marmoset was required to move the cursor from the fixation square to the target rectangle in the force field (Fig. 5b). In each session, blocks of the one-target reaching task both without (nFF block) and with (FF block) the force field were alternatively switched, for one to three times. During 12, 20, and 12 sessions for marmosets A, C, and D, respectively, the marmosets became acclimated to the force field and performed the task without losing motivation. After these sessions, the experimental conditions were fixed (see Methods for details). Immediately after the block was changed from nFF to FF, the cursor trajectory was usually subject to a displacement in the X-axis direction before returning to the target, that is, it performed an "L-turn" (Fig. 5c), as has been observed in mouse and human participants[11,30]. The X-axis displacement averaged over the last ten trials in the preceding nFF (baseline) block was 0.02 ± 0.01 mm ($n = 117$ blocks in 58 sessions from three marmosets), while over the first ten trials of the FF block it was 9.76 ± 0.33 mm (Fig. 5c, d). The X-axis displacement significantly decreased from the first trial to the last trial in the FF block (13.34 ± 0.57 mm vs. 8.40 ± 0.42 mm, $P < 0.01$, Wilcoxon signed-rank test, $n = 117$ blocks in 58 sessions from three marmosets), but did not return to zero (Fig. 5c, d). Immediately after the FF block, a second nFF block (washout block) was performed. In this washout block, the cursor trajectories showed subtle displacements in the opposite direction along the X-axis (Fig. 5c, d; X-axis displacement averaged over the first ten trials of the washout block, −0.36 ± 0.05 mm, $P < 0.01$ compared with the baseline block, Wilcoxon signed-rank test, $n = 117$ blocks in 58 sessions from three marmosets). This "aftereffect" significantly decreased from the first

trial to the last trial in the washout block (−0.97 ± 0.12 mm vs. 0.07 ± 0.05 mm, $P < 0.01$, Wilcoxon signed-rank test, $n = 117$ blocks in 58 sessions from three marmosets). X-axis displacement averaged over the last ten trials of the washout block was 0.002 ± 0.03 mm ($n = 117$ blocks in 58 sessions from three marmosets) and was not significantly different from that over the baseline block ($P = 0.18$, Wilcoxon signed-rank test). These results demonstrate that the head-fixed marmosets learned how to achieve the reward in both the force-field and washout blocks, and showed a weak adaptation to the force field and an after effect of the adaptation.

**Two-photon calcium imaging during task performance**. We conducted two-photon calcium imaging of the right motor cortex of marmosets A and D during task performance. The angle between the cranial window on the primary motor cortex of the marmoset and the horizon was 5–20°, and therefore the optical axis against the cortical surface required tilting for imaging with a high spatial resolution[31]. We therefore introduced a two-photon microscopy technique that allowed tilting of up to 120°. This involved fiber delivery of laser light of 920 nm wavelength, which was directly connected to an x–y scanning box attached to the microscope body (Fig. 6a, b). Emitted fluorescence was captured by a liquid light guide and collected by a photodetector located near the microscope (Fig. 6a, b). The marmoset was set in the chair on the sample stage, and its head was fixed under the objective. The microscope body was tilted 5–20°, to adjust the optical axis so that it was nearly perpendicular to the cranial window (Fig. 6c).

Adeno-associated viruses carrying the tetracycline-controlled transactivator and GCaMP6f genes[4,32] were injected into 4–5 sites of the upper limb of the M1 over 1 month prior to starting the imaging. We conducted two-photon imaging of neuronal somata expressing GCaMP6f during the two-target reaching task (Fig. 7) or force-field adaptation task (the first baseline, first FF, and first washout blocks; Fig. 8). The imaging field was 509 × 509 μm, with the imaging depths from the cortical surface being 250 μm during the former task, and from 120 to 325 μm during the latter task. These depths corresponded to layer 2/3[17,19]. In the two-target reaching task, we conducted two-photon imaging in the same M1 field over six sessions for each marmoset, after the task performance had more or less plateaued (Supplementary Fig. 1). Defining the first imaging session for each marmoset as day 1, and also counting the non-training days, the imaging days were 1, 3, 5, 8, 10, and 12 for marmoset A, and 1, 2, 5, 9, 11, and 12 for marmoset D.

First, we estimated motion artifacts in the imaging data because motion artifacts caused by breathing, heartbeats, and body movement are the biggest problem in two-photon imaging of behaving animals[33]. We corrected motion artifacts using off-line xy-motion correction processing with a finite Fourier transform algorithm[34]. The SDs of x and y shifts during the task performance were less than 2 μm (see Methods for details), which are comparable with those previously reported in mice[33,35]. Neuronal somata with calcium transients were extracted using a constrained non-negative matrix factorization algorithm (CNMF)[36], and motion-corrected traces of the relative fluorescence changes ($\Delta F/F$) in each neuronal soma showed little rapid deflection (Fig. 7a, b; Supplementary Movies 3, 4). Second, we used the CNMF algorithm to define the noise level of $\Delta F/F$ signals as the standard deviation of high-frequency components of $\Delta F/F$ signals in each pixel of the images. The noise level did not depend on the imaging depth, nor did it change across the imaging sessions (Supplementary Fig. 5). Third, we confirmed that the animal did not move its eyes rapidly in conjunction with the

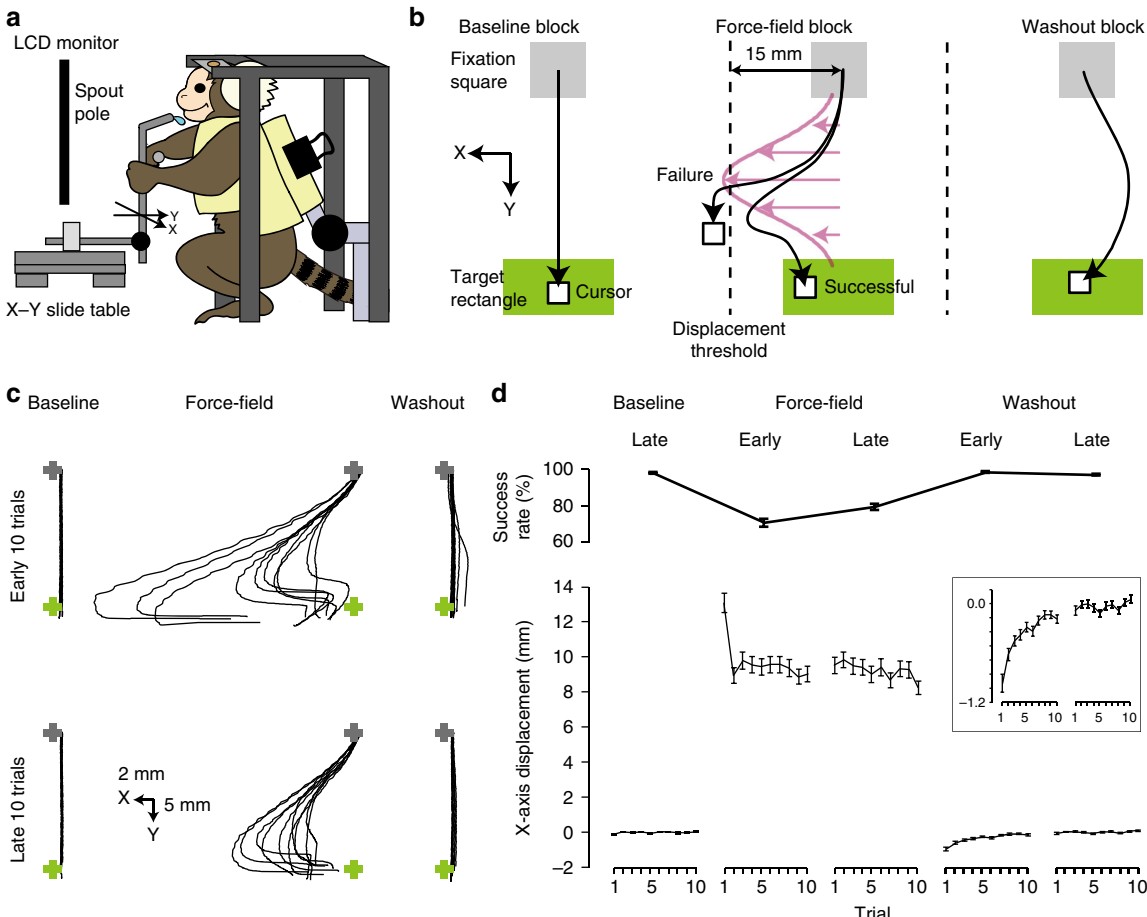

**Fig. 5** Learning of the force-field adaptation task. **a** Scheme of the task apparatus. In this task, the handling pole was replaced with the spout pole and the marmoset manipulated it to move the cursor on the monitor. A fixed pole was put in front of the marmoset's right arm to keep the right arm relaxed. **b** The target rectangle was placed below the fixation square. The width of the target rectangle was twice its height. During the FF block, a velocity- (in Y-axis direction) dependent force field was applied to the pole in the X-axis direction (representing the left direction in the hand workspace). **c** Example of the reaching trajectories in the first baseline, first force-field, and first washout blocks in session 2 from marmoset A. Gray and green crosses indicate the centers of the fixation square and target rectangle, respectively. **d** Success rate averaged over ten trials (top) and X-axis displacement of each trial (bottom) in consecutive baseline, force-field, and washout blocks (n = 117 from three marmosets). The early and late trials were the first ten and final ten reaching trials in each block. Success rate was the percentage of successful trials out of the ten trials. In the inset, the X-axis displacement in the washout block is magnified. Error bars indicate SEM

onset of the cursor movement or the appearance of the target, and that eye fixation was not required at any time (Supplementary Fig. 6).

**Neuronal activity during the two-target reaching task**. After the motion correction, we analyzed the activity in individual neurons during the two-target reaching task (Fig. 7). From the six imaging sessions, 399 active M1 neurons (156 from marmoset A and 243 from marmoset D) and their $\Delta F/F$ traces were extracted using the CNMF algorithm (Fig. 7a, b). First, we examined the spatial relationships in activity between pairs of neurons. Pairwise correlations in activity during successful trial periods (Supplementary Fig. 7a) showed a negative relationship with cellular distance (Supplementary Fig. 7b). This negative relationship was conserved across the training sessions (Supplementary Fig. 7c). The trend was similar during the failure trial periods (Supplementary Fig. 7). These results suggest that neighboring L2/3 neurons in the marmoset motor cortex show correlated activity, as is the case in many mouse cortical areas.

Next, we estimated the activity preferences of individual neurons to the targets 1 and 2 (Fig. 7c). We calculated a direction selectivity index (DSI, see Methods for the calculation) for task-

relevant neurons, which exhibited significantly higher $\Delta F/F$ signals within the time period from $-1$ to $+2$ s from the reaching onset than they did during the fixation period. Approximately half of the active neurons were defined as task-relevant neurons (88 in marmoset A and 127 in marmoset D). DSI values of $+1$ and $-1$ indicate that a neuron responded only during movement towards target 1 or target 2, respectively. The DSI values of the task-relevant neurons from the six imaging sessions were $0.24 \pm 0.06$ ($n = 88$) for marmoset A and $0.09 \pm 0.04$ ($n = 127$) for marmoset D. In both marmosets, the fractions of neurons with a DSI $>0.25$ were significantly higher than those of the shuffled data (Fig. 7d). This may be because both marmosets received much more training on reaching target 1 than target 2 in the one-target reaching or adaptation task, or may also be because the imaging fields were the dominant area for pulling, rather than pushing, the upper limb. The fractions of neurons with a high-selective index value (DSI $>0.5$ or $<-0.5$) were similar across sessions (Fig. 7e). We then examined whether the DSI values of individual neurons were similar across sessions. The task-relevant neurons were automatically pursued across multiple sessions, according to their locations (see Methods). This resulted in 16 neurons in marmoset A and 25 neurons in marmoset D being

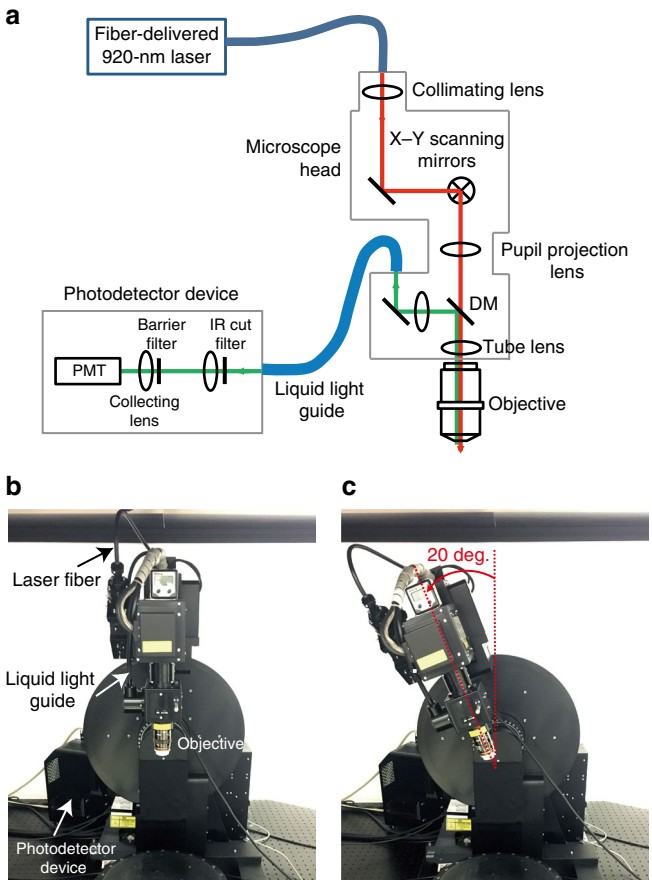

**Fig. 6** Overview of the two-photon microscope for head-fixed marmosets. **a** Scheme of the optical pathway. The laser beam was directly introduced to the microscope head with x–y scanners. Thus, the wavefront of the laser at the exit of the objective was not affected by the tilt of the microscope body. The scanners consisted of a resonant mirror and a Galvano mirror. Emitted fluorescence was spilt by a dichromic mirror (DM; reflection and transmission wavelength ranges, 400–755 nm and 800–1300 nm, respectively), introduced to a liquid light guide, filtered by an IR cutting filter (wavelength range, 400–760 nm; 32BA750 RIF, Olympus) and a barrier filter (FV30-FGR), and then collected by a cooled high-sensitivity photomultiplier tube (PMT). **b** Non-tilt body position of the microscope body. **c** Body position with a tilt angle of 20° around the front-to-back axis

identified as task relevant in multiple sessions. Correlations in the DSI values of these individual neurons between sessions were significant in both marmosets, irrespective of the time interval (≤5 days or >5 days; Fig. 7f). These results indicate that the L2/3 motor cortical neurons with a strong activity preference for the direction of upper-limb movement on a particular day tended to maintain this preference across days.

**Neuronal activity during the adaptation task.** During motor adaptation, individual cortical neurons in the macaque monkey show dynamic changes in activity within 10–30 min[37–39]. We tested whether two-photon calcium imaging detected activity changes in individual neurons of the marmoset during the force-field adaptation task. From seven sessions in marmoset A, 113 active neurons were extracted, and from eight sessions in marmoset D, 425 active neurons were extracted. Approximately 40–50% of these neurons were task relevant (56 in marmoset A and 163 in marmoset D; Fig. 8a). Of the task-relevant neurons, 60–70% (38 neurons in marmoset A and 98 neurons in marmoset D) showed significant changes in activity between the three

blocks (Kruskal-Wallis test). Then, these neurons were categorized into seven groups according to the difference in the mean task-relevant activity between pairs of blocks (Dunn-Sidak test; Fig. 8b). Neurons with different mean activity between the baseline and FF blocks and between the FF and washout blocks, but not between the baseline and washout blocks, might reflect large displacement of the cursor trajectory in the FF block (neurons A1 and D1 in Fig. 8a). Neurons with different mean activity between the baseline and washout blocks probably changed the activity despite the similar cursor trajectory between the baseline and washout blocks (neurons A2 and D2 in Fig. 8a). The fraction of neurons in each group was relatively similar between marmosets (Fig. 8b), and the fractions of almost all groups were significantly higher than those calculated from trial shuffled data (the 95th percentile of the shuffled values in both marmosets; Fig. 8b). Therefore, the force-field adaptation task rapidly induced a variety of changes in the activity of the L2/3 motor cortical neurons of the marmosets.

**Two-photon calcium imaging of dendrites and axons.** Another advantage of two-photon imaging is its subcellular resolution. We successfully imaged dendritic compartments in layer 1 of M1 while the marmoset performed nFF blocks of the force-field adaptation task (Fig. 9a). In 9 out of 15 imaging fields, motion-corrected $\Delta F/F$ traces of some dendritic compartments showed task-relevant activity (Fig. 9b, c). In pixel-based correlation maps of the imaging field, the morphology of several dendritic branches was represented by pixels with high correlation coefficients, probably reflecting dendritic branches originating from the same neurons (Fig. 9d). Similarly, we were able to monitor the task-relevant activity of single axonal boutons in layer 1 of M1 in one out of three imaging fields (Fig. 9e–h). These results demonstrate that two-photon calcium imaging with a subcellular resolution is also feasible in the cerebral cortex of head-fixed marmosets performing an upper-limb movement task.

## Discussion

The major technical advances in this study are the establishment of a stable restraint for the marmoset trunk using a jacket, and the step-by-step protocols to train the head-fixed and trunk-restrained marmoset to perform multiple upper-limb movement tasks. We have summarized the caveats for each step of the task protocols in Table 1. The training durations for the tasks without head fixation varied slightly across the marmosets, but all marmosets finally performed the tasks (Supplementary Fig. 1). The across-animal differences may be due to differences in the motivation to obtain a reward, as well as differences in the learning process. In head-fixed marmosets, differences in task performance improvement between individuals have also been reported for other tasks, such as tone detection and eye fixation tasks[25,26]. Thus, compared with macaques, it appears to be more important to optimize the task parameters and schedule for each marmoset, especially during the initial training step for motor tasks with head fixation. For marmosets A and D, 4–5 months were required for them to skillfully perform the two-target reaching task. This duration was similar to that required to train macaques to perform stably an eight-direction cursor movement task with force-field adaptation, which is more difficult than the one- or two-target reaching tasks[40]. During the adaptation task, the marmoset did not return the cursor trajectory to a straight one in the FF block (Fig. 5), in contrast to previous studies on humans and macaques. This may be because the number of trials within a session was smaller than those used in other primate studies (e.g., approximately 160 successful trials in both FF and nFF blocks[40]). This could be overcome by reducing the volume of

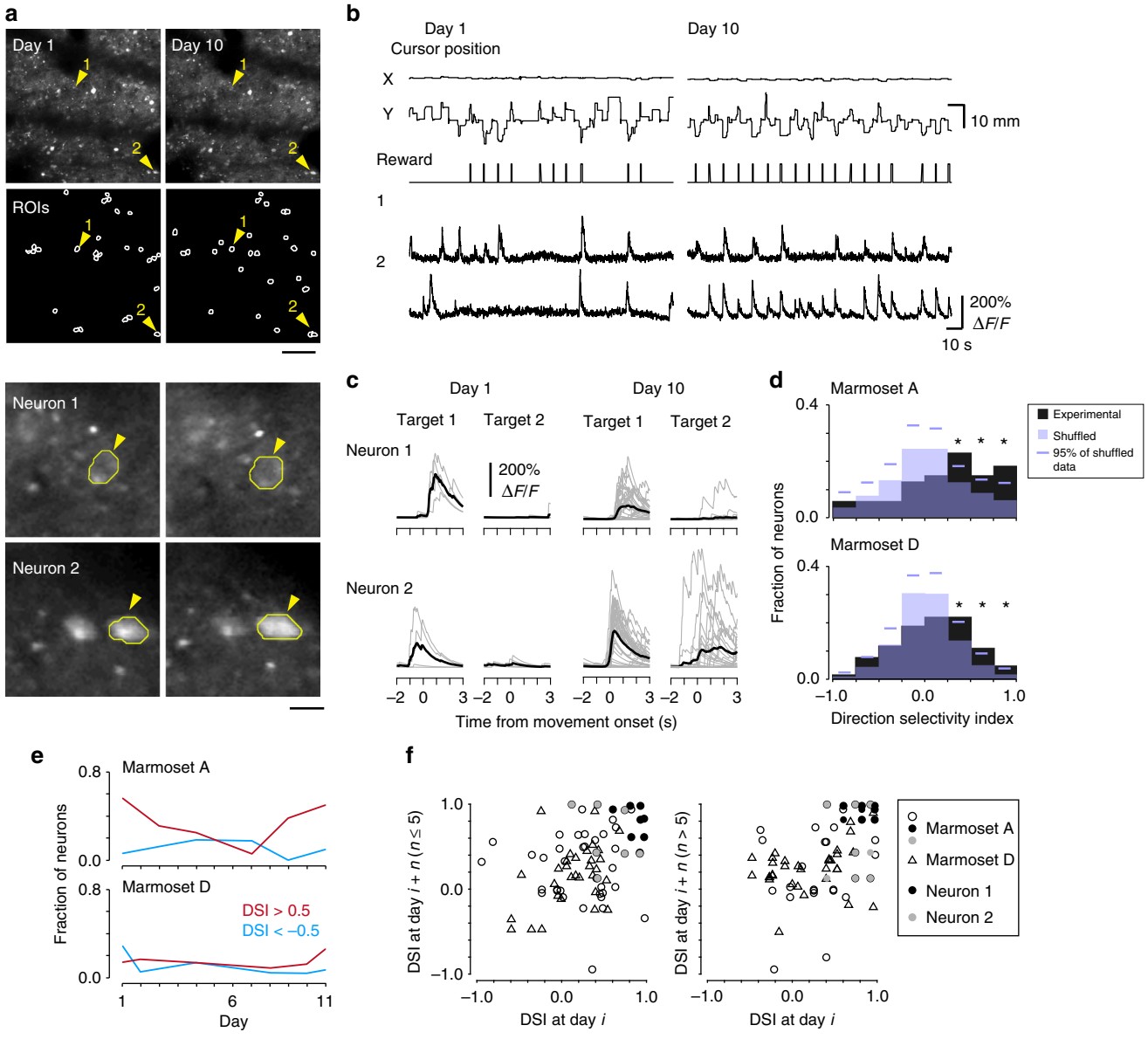

**Fig. 7** In vivo two-photon multi-day imaging in the motor cortex during the reaching task. **a** Time-averaged two-photon images of GCaMP6f from marmoset A on imaging days 1 and 10 (first row), and ROIs identified by the CNMF algorithm (second row). Magnified areas around neurons 1 (third row) and 2 (fourth row), and their contours, are also shown. Scale bars: 100 μm for the whole images and 15 μm for the magnified images. **b** X and Y positions of the cursor, reward timing, and the traces of motion-corrected raw fluorescence signals in two representative neurons from imaging days 1 and 10 shown in **a**. **c** Traces of denoised $\Delta F/F$ signals of neurons 1 and 2 aligned to the cursor movement onset. Gray and black traces represent individual trials and the average, respectively. **d** Histogram of DSI of neurons pooled from six imaging sessions (black boxes). Purple boxes and lines indicate the distributions of shuffling-averaged DSI from the trial shuffled data, and the 95th percentile of 1000-time shuffling in individual bins, respectively. For both marmosets, fractions in the three bins with >0.25 DSI were above the 95th values (*$P < 0.05$). **e** Fractions of neurons with DSI >0.5 (red) and <−0.5 (cyan) in each session. CCs between the fraction of the neurons with DSI >0.5 and the imaging day, −0.09 and −0.03, $P = 0.91$ and $P = 1.0$, in marmosets A and D, respectively; neurons with DSI <−0.5, −0.14 and −0.54, $P = 0.80$ and $P = 0.29$, in marmosets A and D, respectively. **f** Similarity in the DSI of the same task-relevant neurons between different imaging days. Each point represents the DSIs of the same neuron on an imaging day and a following day ≤5 days apart (left) or >5 days apart (right). The CCs for the DSIs with session intervals ≤5 days were 0.23 and 0.45 ($n = 50$ and 31, $P < 0.05$ for both cases) for marmosets A and D, respectively, while those for an interval >5 days were 0.45 and 0.30 ($n = 33$ and 35, $P < 0.01$ and $P < 0.05$), respectively

the reward in each trial. With further improvements in the tasks and slightly longer training sessions, the marmoset may be able to learn very similar upper-limb movement tasks to those performed by other primates. The marmoset appears to be an appropriate non-human primate model for research on motor control.

To make full use of the newly-developed protocols, we conducted two-photon calcium imaging of the motor cortex during performance of the motor tasks. In the two-target reaching task, we found that the extent of direction preferences in each imaging field was similar across days, and that some neurons showed high direction preferences across days. This is consistent with results from macaques showing that spiking activity and local field potentials representing movement are stable across different recording days[41,42], although single-unit activity can stably record the same neuron in the macaque premotor cortex for two days[42]. Electrical units are generally recorded from deep layers, whereas

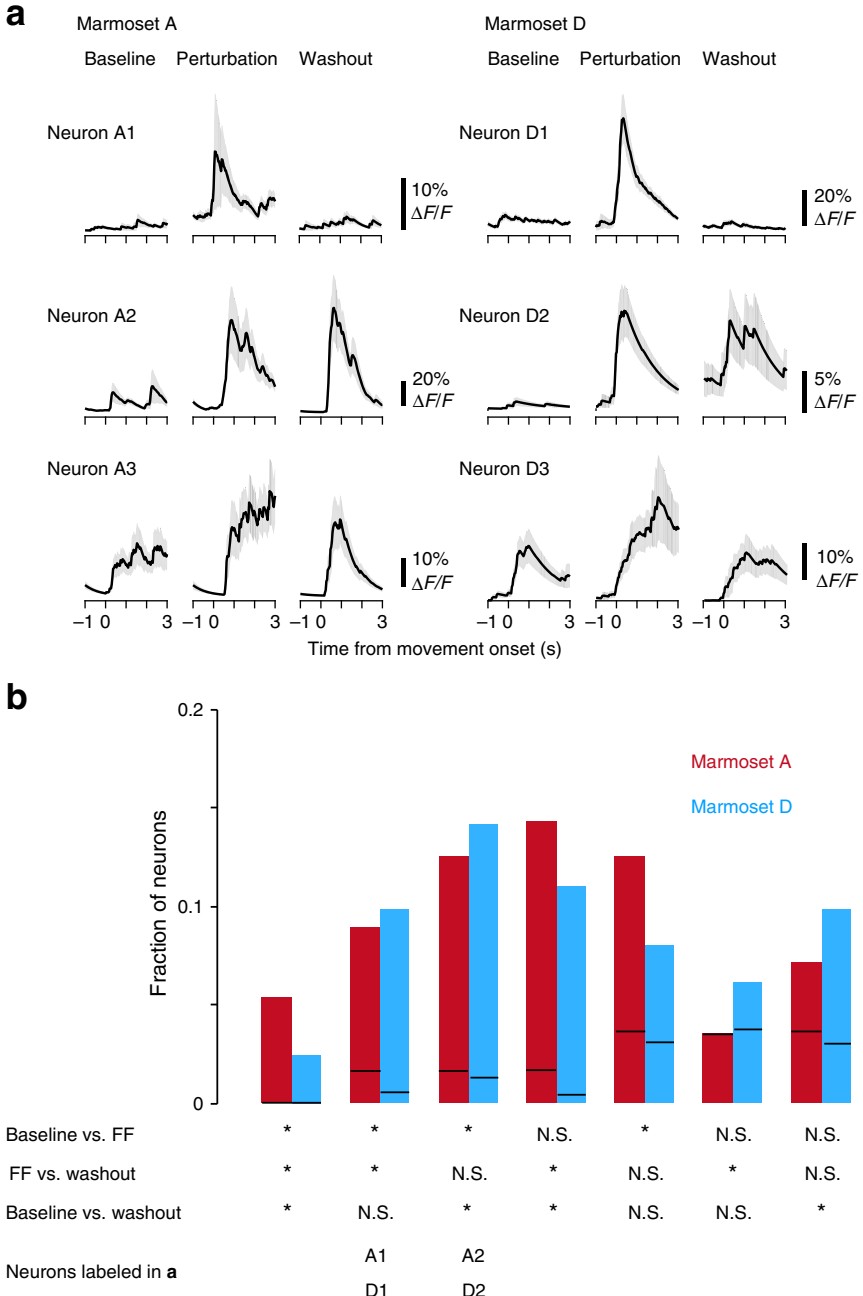

**Fig. 8** Changes in neuronal activity during the force-field adaptation task. **a** The task-relevant activity of six representative neurons (three from marmoset A and three from marmoset D). Denoised $\Delta F/F$ traces were aligned to the cursor movement onset. Lines and shaded areas represent mean ± SEM ($n =$ 12–20 successful trials occurring during imaging for each block). Neurons A3 and D3 did not show significant differences in activity between the three blocks (Kruskal-Wallis test, $P = 0.53$ and $0.14$, respectively). **b** Fractions of seven groups of neurons that showed significantly different activity between the three blocks (Kruskal-Wallis test, $P < 0.05$) with respect to the task-relevant neurons for each marmoset. According to the significance in the post-hoc test ($P < 1 - [1 - 0.05]^{1/3}$), these neurons were classified into seven groups. Red, marmoset A. Cyan, marmoset D. Asterisk indicates a statistical difference between the pair of blocks shown on the left. N.S. indicates a non-statistical difference. The bottom indicates which groups of neurons (A1, A2, D1, and D2) in **a** were assigned to. Black line in each fraction box indicates the 95th percentile of the fraction obtained from trial shuffled data. Fractions, except for that in the second row from the right in marmoset A, were larger than the corresponding 95th percentiles

we imaged superficial layers. The superficial layers of the motor cortex receive strong signals from the sensory cortex[30,43], while layer 5 neurons include corticospinal neurons, which are directly involved with muscle activity. Thus, the superficial and deep layers may show different neuronal activity and plasticity, as we previously suggested for the mouse motor cortex[3]. In the force-field adaptation task, we found that some neurons in the local area changed their mean task-relevant activity during the

perturbation block, while others did not significantly change the mean task-relevant activity, irrespective of the perturbation. Previous electrophysiological studies showed that individual neurons in the macaque dynamically changed their activity during the performance of motor adaptation tasks[37–39]. One of the next challenges is to apply two-photon imaging of red GECIs or three-photon imaging[44,45] to layer 5 (at depths >1 mm from the cortical surface) of the marmoset, and compare the neuronal

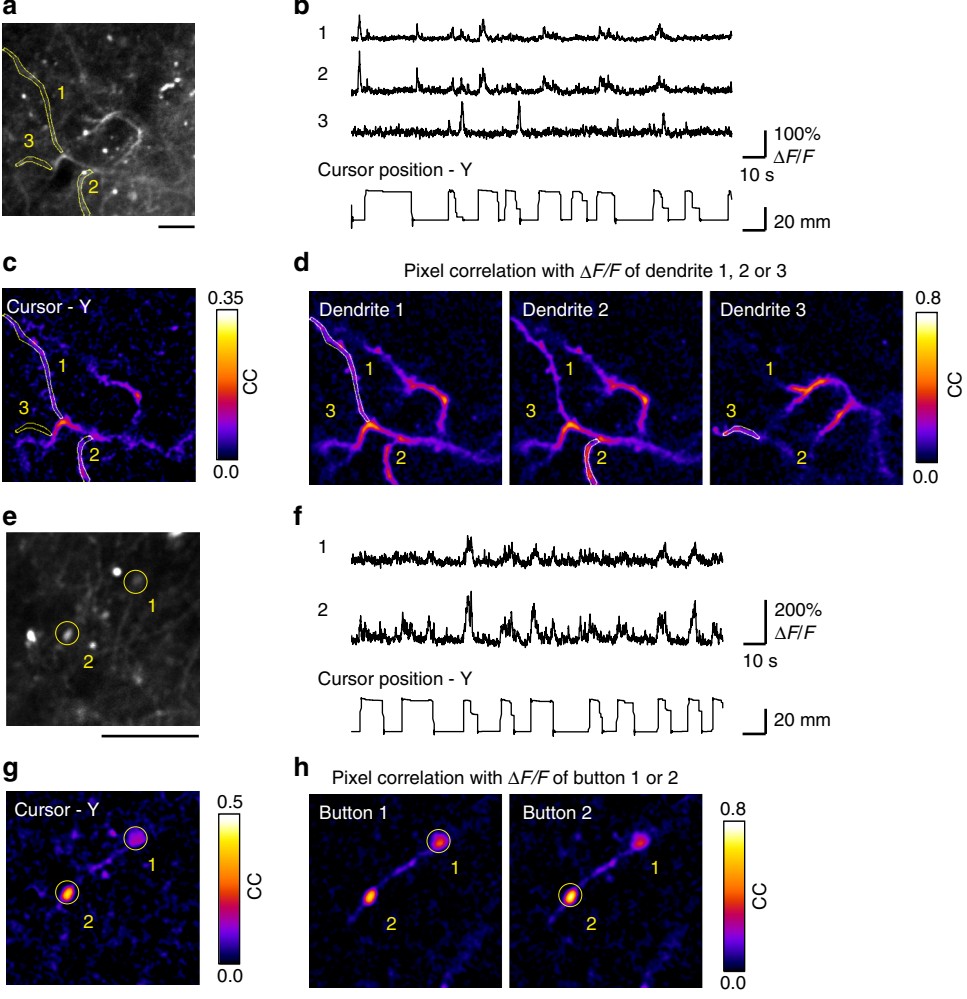

**Fig. 9** Two-photon imaging of subcellular activity in layer 1 of the M1 during the task. **a** Representative time-averaged two-photon image of dendritic compartments at a depth of 51 μm from the cortical surface in the M1 of marmoset D. Scale bar, 30 μm. **b** Motion-corrected ΔF/F traces of three representative dendritic compartments indicated by the numbers in **a**. The Y-axis trajectory of the pole is shown at the bottom. **c** Map of correlation coefficients between the Y-axis cursor trajectory and the ΔF/F trace in each pixel. **d** Map of correlation coefficients in the ΔF/F traces with ROI 1 (left), ROI 2 (middle), and ROI 3 (right). **e** Representative time-averaged two-photon image of axonal buttons at a depth of 46 μm in the M1 of marmoset D. Scale bar, 20 μm. Other conventions are the same as in **a**. **f** Motion-corrected ΔF/F traces of two representative axonal buttons indicated by the numbers in **e**. **g** Map of correlation coefficients between the Y-axis cursor trajectory and the ΔF/F trace in each pixel. **h** Maps of correlation coefficients for the fluorescence changes for ROI 1 (left), and ROI 2 (right)

activity between two-photon imaging and electrical recording. If the neuronal activities in the L2/3 and L5 neurons of M1 and the input axons carrying the sensory and/or sensory error signals can be imaged, our knowledge on the motor adaptation circuit, which was acquired by electrophysiology, should be advanced[39,46,47].

Research on non-human primates allows us to examine both motor and cognitive processes with more relevance for the human brain than examining the processes in non-primate species. Non-head-fixed marmosets show cognitive functions such as decision making, working memory, and attention shifting[48–52]. We demonstrated that even head-fixed marmosets were able to wait several seconds for the visual cue to appear before moving the cursor. Thus, we expect that head-fixed marmosets could learn to perform many cognitive tasks with a delay period. In addition, the marmoset is more prosocial, and its vocal communication is more developed than that of the macaque[53–55]. Social tasks for head-fixed marmosets should also therefore be achievable[56,57]. Furthermore, it may be possible to estimate abnormalities in population dynamics in a Parkinson's disease marmoset model if the force-field adaptation task in the present

study is modified for use by individuals with the disease[58]. Transgenic disease models of the marmoset have already been developed[59–61]. Two-photon calcium imaging in the behaving marmoset could open a new era of understanding of the neuronal dynamics relevant to higher brain functions, and present new insights into psychiatric and neurological diseases in the human.

Tasks that involve the forelimb, such as lever-pull/push tasks and force field-based adaptation tasks, are well established in head-fixed rodents, with the learning processes having been examined[2,3,29,30,35,62]. Thus, changes in neuronal activity during long-term learning or short-term adaptation in similar motor tasks can be compared between the mouse and marmoset. We demonstrated that two-photon calcium imaging can resolve dendritic and axonal activity in the marmoset, as has been shown in the mouse[4,63,64]. It may be possible to determine whether nonlinear dendritic computation is critical for higher brain functions in non-human primates[65,66]. The techniques developed in the present study have the potential to connect findings from rodent and primate research, which will help our understanding

**Table 1 A list of caveats for each step of the tasks**

| Task | Caveat |
| --- | --- |
| Habituation | If the marmoset shows any agitated behavior, the training is stopped. |
| Dish- or spout pole-pull task | Dish or spout pole is placed at a location whose distance from the marmoset is equivalent to that of the maximum reach of the marmoset. If motivation is not apparent, the food is changed. |
| Self-initiated pole-pull task without head fixation | Repeat this step until the marmoset concentrates on the task and does not move their head too much over several sessions. |
| Visually cued pole-pull task | The penalty extending the blank period is critical in training the marmoset not to pull the pole during the blank period. |
| One-target reaching task | The holding time within the target square should be gradually increased as the training progresses. |
| Two-target reaching task | Increase the proportion of target 2 trials in early sessions. |
| Force-field adaptation task | If the marmoset stops the task during the FF block, the velocity-dependent force is reduced. |

of the evolutionary mechanisms through which higher motor and cognitive functions have been achieved.

## Methods

**Animals**. All experiments were approved by the Institutional Animal Care and Use Committee of the National Institutes of Natural Sciences, and the Animal Experimental Committee of the University of Tokyo. Four laboratory-bred adult male common marmosets (*Callithrix jacchus*; marmosets A, B, C, and D) were trained on the behavioral tasks described in the present study. The marmosets were 37–52 months old (weight, 318–355 g) when the habituation started. Another marmoset aged 12 months (weight, 320 g) was used for the intracortical micro-stimulation (ICMS) experiment to identify the motor cortex. All marmosets were kept on a 12:12 h light-dark cycle and were not used for other experiments prior to the present study.

**Virus production**. Production procedures for vector plasmids, pAAV-Thy1S promoter-tetracycline-controlled transactivator 2 (tTA2) and pAAV-TRE3G promoter-GCaMP6f-WPRE, followed those described previously[21]. This tetracycline-inducible gene expression system was used as it amplifies GECI expression sufficiently enough to allow fluorescence changes to be detected in the anesthetized marmoset brain[21]. AAV vectors were also produced as described previously[21].

**Surgical procedures**. All surgical procedures and AAV injections were performed under aseptic conditions[21]. Marmosets were placed in a stereotaxic instrument (SR-5C-HT; Narishige, Tokyo, Japan) with anesthesia maintained using inhalation of isoflurane (1.5–4.0% in oxygen). Pulse oxygen (SpO2), heart rate, and rectal temperature were continuously monitored to judge the marmoset's condition. Cefovecin (14 mg kg$^{-1}$) as an antibiotic drug, and carprofen (3.75 mg kg$^{-1}$) as an anti-inflammatory drug to reduce pain and inflammation during and after surgery, were administered intramuscularly. Acetated Ringer's solution (10 ml) including riboflavin sodium phosphate (200 μg) was also administered subcutaneously. After hair removal by a depilatory and sterilization with povidone iodine, the skull was exposed. Lidocaine jelly was applied to wound sites to reduce pain.

ICMS was conducted on one marmoset, to localize the motor cortex. After craniotomy had been performed, the underlying dura mater was removed, and seven small screws were anchored to the skull. A universal primer (Tokuyama, Tokyo, Japan) was applied to the surface of the skull, and a head plate (CFR-1; Narishige) was attached to the skull using dual-cured adhesive resin cement (Bistite II; Tokuyama). The head plate was then fixed to an apparatus. After reduction of the isoflurane anesthesia, a tungsten microelectrode with an impedance of 0.5 MΩ (World Precision Instruments, FL, USA) was inserted to a depth of 0.8–1.6 mm from the cortical surface, and a train of 20 cathodal pulses (0.5 ms duration at 200 Hz) were applied. ICMS of 7–32 μA at 9–10 mm anterior from the interaural line and 4–5 mm lateral from the midline induced upper-limb movement. This result was consistent with those of other studies[16,17,19]. Thus, we considered this area to be the primary motor cortex (M1). For virus injections, a 4.5 mm diameter circular craniotomy was made over the right M1 and the dura mater was removed. A pulled glass pipette (broken and beveled to 60–70 μm outer diameter; Sutter Instruments, CA, USA) and a 5 μl Hamilton syringe were back-filled with mineral oil (Nacalai Tesque, Kyoto, Japan) and front-loaded with virus solution. In marmosets A and D, a mixture of rAAV2/1-Thy1S promoter-tTA2 and rAAV2/1-TRE3G promoter-GCaMP6f-WPRE was injected at 0.1 μl min$^{-1}$ for 5–10 min with a syringe pump (KDS310; KD Scientific, MA, USA). The injection sites were 0.5–1 mm apart from each other. The viral preparations were adjusted to the final concentration of 0.17–0.20 × 10$^{12}$ vector genomes ml$^{-1}$ for rAAV2/1-Thy1S promoter-tTA2, and 1.0 × 10$^{12}$ vector genomes ml$^{-1}$ for rAAV2/1-TRE3G promoter-GCaMP6f-WPRE. The pipette was inserted vertically approximately 500 μm ventral from the brain surface. After injection, the pipette was maintained in place for an additional 5–10 min, before being slowly withdrawn. A window consisting of a 5.5 mm circular glass coverslip (approximately 100 μm thickness; Matsunami Glass, Osaka, Japan) cemented to four sheets of 3 mm circular glass coverslips (approximately 300 μm

thickness; Matsunami Glass) with UV-curing optical adhesive (NOR-61; Norland Optical Adhesive, NJ, USA) was pressed onto the brain surface, and the edge was sealed with dental cement (Fuji Lute BC; GC, Tokyo, Japan) and dental adhesive resin cement (Super bond; Sun Medical, Shiga, Japan). The head plate was attached to the skull as described above.

**Task apparatus**. The task apparatus consisted of a marmoset restrainer, a head plate holder for head fixation (O'Hara & Co., Tokyo, Japan), an aluminum feeding tube (inner diameter of 2 mm and outer diameter of 6 mm) connected to a custom built syringe pump, a 7 inch LCD monitor (LTM07C382J, 1024 × 600 pixels; Toshiba, Japan) placed 10–15 cm in front of the animal's head, and a single/dual-axis manipulandum. Marmosets wearing a jacket[27] made of cotton were restricted to the restrainer by a metal-pole supporting arm (diameter of 10 mm) attached to the plastic base plate of the restrainer. The supporting arm was inserted into the sleeve of the jacket[27]. For tight restriction, the sleeve was fastened with one or two clips. This jacket was introduced because we expected that the restraint provided by the jacket would allow the marmosets to adopt a more comfortable posture in the chair than would be attainable using a solid tube or plate. A reward drop of apple juice (50–200 μl) was delivered from a feeding tube positioned near the mouth of the marmosets by pressure from a syringe pump. The manipulandum, which consisted of a stand pole (diameter of 6 mm, stainless steel), an L-shaped spout pole, or a metal dish, was attached to a linear slider (that could move 35 mm along a front-to-back axis) or an X–Y slide table (53 × 90 mm workspace for X and Y, respectively). A robotic arm (Geomagic touch; 3D Systems, NC, USA) was connected to the slider, and the position of the stand pole or the L-shape pole, and the force applied to them, were monitored. In some experiments, licking behavior was monitored with a CCD camera (DMK 33GP1300; Imaging Source, Taipei, Taiwan). The digital and analog signals were controlled by custom-made software written in LabView (National Instruments, TX, USA) and Visual C++ 2008 (Microsoft, WA, USA), and the analog data were archived by an NI-DAQ device (National Instruments). The analog data were sampled at 1 kHz, while the position of the pole and the force generated by the robotic arm were sampled at 40 Hz or 1 kHz.

**Behavioral tasks**. The marmosets were trained for <1 h per day (between 09:00 and 19:00), for 1–5 days per week. On training days, food and water were reduced to maintain body weight at approximately 90% of normal weight. On non-training days, body weights fully recovered. Each week, the jacket was put on the marmoset when the first session started and was removed when the final session ended. The training schedule for each of the four marmosets is shown in Supplementary Fig. 1.

**Training protocols prior to the visually cued pole-pull task**. Step 1: Habituation to body restraint in the restrainer: To restrain the marmosets' bodies and fix their heads during behavioral tasks, they were habituated to the body restraint in the restrainer. Marmosets were placed in front of a food bowl and a water bottle (Fig. 1b), and allowed to freely access food and water. Initially, marmosets tended to take off the jacket immediately after the task was started. In this case, additional food (e.g., marshmallow and a piece of cookie) was given to keep the marmoset in the restrainer. When marmosets demonstrated agitated behavior, even with the additional food, the training was stopped and the animals were returned to their home cage. This step was repeated until they stayed calm in the restrainer for more than 5 min.

Step 2: Dish- or spout pole-pull task: The marmosets were then trained to retrieve a food pellet that was placed on an aluminum dish attached to the linear slider. Experimenters put a food pellet on the dish and set it far from the animal's body (default position), with the marmosets being required to pull the edge of the dish to obtain the pellet. Alternatively, the animals were trained to pull an L-shape spout pole attached to the slider by 15–20 mm, to allow them to lick a drop of apple juice (70–200 μl) delivered to the tip of the pole. After the marmosets retrieved the pellet from the dish or licked the tip of the pole, the dish or pole was returned to the default position by the experimenters, who then observed whether or not the marmoset again pulled the empty dish or pole in an attempt to obtain another

reward. This step was repeated until the experimenters recognized that the marmosets came to volitionally pull the dish or pole.

Step 3: Self-initiated pole-pull task without head fixation: The dish or spout pole was replaced with a metal pole, and the marmosets were trained to volitionally pull the pole with the left hand to obtain a reward (self-initiated pole-pull task). After the pole was pulled, a juice drop was given from the tip of a feeding tube kept near to the marmosets' mouth. The marmosets soon started to pull the pole and lick the tip of the tube when the reward was delivered. Each session was terminated when the marmoset stopped sitting quietly in the chair or the total amount of juice given was approximately 20 ml. After several sessions, the animals normally kept their mouth very near to the feeding tube and pulled the pole with little head movement.

Step 4: Self-initiated pole-pull task with head fixation: After step 3, the animals' head was fixed by clamping the implanted metal plate in a head plate holder (Fig. 1d). For marmosets A and B, a spring force was applied to the pole to direct it to the default position 1–2 s after the pole was moved beyond the threshold position (located 15–20 mm from the default position). For marmosets C and D, the spring force was applied throughout the procedure. The spring force was $k \times d$, where $k$ is the spring force constant set to 0.1–0.2 N mm$^{-1}$, and $d$ is the distance (in mm) between the default position and the position of the pole.

**Visually cued pole-pull task**. After the head-fixed marmosets had learned the self-initiated pole-pull task, they were all trained to perform a visually cued pole-pull task in which they had to pull the pole when a cue was presented on a LCD monitor. The task consisted of cued and blank periods. During the cued period, a white square (40 mm each side, intensity of 63.1 lux) was presented as the cue. The vertical position of the cue on the monitor indicated the position of the pole in the back-to-front axis (a 10 mm pole-pull movement corresponded to a 20 mm upward movement of the cursor). The color of the cue was changed to green (25.0 lux) when the pole position exceeded the threshold position (Fig. 2b). In the blank period, marmosets needed to keep the pole below the threshold position. Every time the pole was pulled beyond the threshold during this blank period, the duration of this period was extended by $t$ ms, where $t$ was the duration in which the pole position exceeded the threshold. Initially, the blank period was not included, so that the task was the same as the self-initiated pole-pull task, except for the displaying of the cue on the monitor. After the initial training, the duration of each cued period was set to 12.5–10 s and the blank periods to 2.5–5 s. After 3–21 sessions, the duration was further shortened: each cued period was 3 or 2 s and each blank period was 3 or 4 s (resulting in a cycle of approximately 6 s for a cued period and a blank period). For marmosets A, B, and C, this duration time was fixed for the next seven sessions. For the first five of these seven sessions, the spring force constant $k$ was set to 0.6–0.8 N mm$^{-1}$ during the blank period, to train the marmosets not to pull the pole. For the remaining two sessions, $k$ was set to 0.15–0.2 N mm$^{-1}$ throughout both periods. In these early sessions, the threshold position was adjusted in the range of 15–20 mm. Finally, the threshold position and duration of the cued period were 15 mm and 3 s, respectively, for all three of the marmosets. The blank period was randomly varied between 3–4 s for each trial in these sessions. Marmoset A had a lower hit rate than the other marmosets in sessions 1–11 (Fig. 2f), because of a larger number of false alarms (Fig. 2e), although this across-animal difference reduced in the later training sessions. After this task training, the training of marmoset B in other tasks was aborted due to its poor physical condition. Marmoset D was trained for five sessions with the latter protocol for the visually cued pole-pull task, commencing after the other marmosets had learned all the tasks, including those described below. As the main purpose of marmoset D was to perform the force-field adaptation task, marmoset D ended this visually cued pole-pull task in session 8, with its performance still having not reached a plateau. Thus, the results from marmoset D are not included in Fig. 3.

**One-target reaching task**. Marmosets A and C were trained to control the 2D manipulandum to move a white cursor (a square, 10 mm each side) from a gray fixation square (20 mm each side, intensity of 30.2 lux) to a green target square (20 mm each side, intensity of 25.0 lux) on the LCD monitor, and to then hold it inside the target square. The position of the cursor represented that of the pole attached to the X-Y slide table (53 × 90 mm hand workspace). A pole movement of 10 mm corresponded to cursor movement of 20 mm. The vertical location of the center of the fixation square was at the level of the marmoset's eye. The target center was located 30–50 mm straight below the fixation center on the monitor, corresponding to a movement of 15–25 mm towards the marmoset in the workspace. The horizontal locations of the squares were modified to be in front of the marmoset's face.

Each trial of the reaching task consisted of a fixation period and a reaching period followed by an inter-trial interval (ITI). The fixation and target squares were presented on the monitor during the fixation and reaching periods, respectively. During the fixation period, the marmoset had to move the cursor close to the fixation square. When the pole entered a circular area with a diameter of 10–20 mm of the center of the fixation square, a spring force was applied to the pole to move it to the default position (i.e., to move the cursor to the fixation center). The spring force constant $k$ was set to 0.5–1.5 N mm$^{-1}$, and $d$ was calculated as the distance (in mm) between the cursor and the center of the fixation square in the workspace. The marmosets needed to hold the pole with a spring force <0.8–1.5 N (corresponding to 0.53–3.0 mm from the center) for 0.7–1.0 s to finish the fixation

period. Every time a pole was moved with a spring force >0.8–1.5 N, the duration of this period was extended by 0.1–0.2 s. The reaching period then started (the target square appeared) after the fixation period ended (the fixation square disappeared). During this period, the marmosets had to move the cursor to the target square and hold it within the target square for 10–300 ms (holding time) to obtain a juice reward (successful trial). When marmosets did not hold the cursor within the target square, the trial was terminated (failed trial). The reaching period ended (the target square disappeared) 2 s after the reward was given, or 0.05 s after a failed trial was terminated. The durations of the ITI were 1.5 s and 3.5 s after successful and failed trials, respectively. No force was applied during the reaching periods and ITIs. To train the marmoset to perform this task, the holding time was gradually increased as the training progressed. From session 28 for marmoset A and 16 for marmoset C, the holding time was fixed to 300 ms.

The straightness of the cursor trajectory was assessed according to a straightness index (SI), which was the length between the start and end points of the cursor ($L_o$) divided by the length of its actual trajectory ($L_{path}$). The trial-to-trial variability of the cursor trajectory for each session was defined as the mean of the root mean square deviations (RMSDs) of the X/Y coordinates of individual trajectories from those of the trial-averaged trajectory. For each trial, RMSD was calculated as

$$\sqrt{\frac{1}{n}\sum_t^n (x_t - \bar{x}_t)^2},$$ where $n$ is the number of time points during the period from −100 ms to +500 ms of the cursor movement onset, and $x_t$ and $\bar{x}_t$ are the X/Y coordinates of the trajectory in the trial and the trial-averaged trajectory at time point $t$, respectively.

Marmoset D performed this task using the pole for 3 days, and a spout pole (see below) for 2 days. Marmoset D then started the force-field adaptation task before its performance in the one-target reaching task had plateaued. Thus, the results from marmoset D are not included in Fig. 3.

**Two-target reaching task**. Marmosets A and D were trained on the two-target reaching task. This was similar to the one-target reaching task (see above), but with the target square (25.0 lux) being displayed straight above or below the fixation square (30.2 lux) on the monitor. In this task, the size of the target and fixation squares on the monitor was 8 × 8 mm, which corresponded to an 8 × 8 mm hand workspace. The size of the cursor was 3 × 3 mm, and the vertical location of the center of the fixation square was 5 mm above the vertical position of the marmoset's eye. Trials were terminated when the cursor was moved more than 16 mm from the center of the target. In the first four sessions for marmoset A, the target was switched after each successful trial. In the following sessions, the target was displayed at random for each trial. As marmoset D had already been trained to perform the adaptation task for approximately 30 sessions, the marmoset preferred to move the pole towards the direction of target 1. To habituate it to move towards target 2 more frequently, the probability of a target 2 representation was increased within the range of 60–100% for the first nine sessions. In the following sessions, the probability was fixed to 50%.

**Force-field adaptation task**. Marmosets A, C, and D were trained to perform the force-field adaptation task. In this task, the handling pole was replaced with an L-shape spout pole, similar to the spout lever used in a volitional lever-pull/push task for rats[29]. The position of the spout pole was adjusted so that the marmosets could lick the tip of the pole when they pulled the pole to move the cursor to the target rectangle. The width and height of the target rectangle (25.0 lux) were 16 and 8 mm, respectively, corresponding to a 16 × 8 mm hand workspace. The vertical location of the center of the target rectangle was 5 mm below the vertical position of the marmoset's eye. The fixation center was positioned above the target center (27 mm above for marmoset A, 31 mm above for marmoset C, and 25.5 mm above for marmoset D). A trial was defined as successful when the marmoset moved the cursor to the target rectangle and held it there for 300 ms, within the 800–1000 ms period after the movement onset, or it was otherwise defined as a failed trial. Trials were terminated when X-axis displacement of the pole exceeded 15 mm. Training started with a block of 40–120 successful trials without a force field (baseline block), followed by a block with 20 successful trials with a force field (FF block, the total number of trials involving failures ranged from 20 to 57), and a block with 40 successful trials without a force field (washout block). In the FF block, a velocity-dependent force field was applied to the pole[67]. The spring force in the X direction at each time point was calculated as $k_V \times V_Y$, where $k_V$ was a velocity-dependent force constant set to 20.0 N ms mm$^{-1}$, and $V_Y$ was the velocity of the cursor in the Y direction (mm ms$^{-1}$). $V_Y$ at time point $t$ (ms) was calculated as $[Y(t)-Y(t-10)]/10$, where $Y(t)$ indicates the Y-position of the cursor at $t$. No force was applied in the baseline and washout blocks. After the washout block ended, the FF block and washout block were repeated, until the marmosets stopped moving the pole or 20 ml of the reward juice had been given. In the first 12, 20, and 12 sessions for marmosets A, C, and D, respectively, the displacement threshold for trial termination was modified within the range of 15–30 mm, and the velocity-dependent force constant was adjusted to 1.0–20.0 N ms mm$^{-1}$. In these sessions, the FF block was also extended or shortened, until the marmosets had succeeded for 10–100 trials. After these sessions, the experimental conditions were fixed.

**Two-photon imaging**. Imaging was conducted with a custom-built two-photon microscope (Olympus, Tokyo, Japan) equipped with a water immersion objective lens (for imaging of neuronal somata: Olympus XLPLN10XSVMP, numerical aperture of 0.6, working distance of 8 mm; for imaging of dendrites and axons: Olympus XLSLPLN25XSVMP2, numerical aperture of 0.95, working distance of 8 mm) and an Nd-based fiber-delivered femtosecond laser (Femtolite FD/J-FD-500, pulse width of 191–194 fs, repetition rate of 51 MHz; IMRA, MI, USA) at a wavelength of 920 nm. In our previous study, in which two-photon calcium imaging of cortical neurons was conducted in anesthetized marmosets, this was solved by tilting the marmoset chair using a goniometer-equipped rotating stage[21]. However, in the awake behaving marmosets, tilting the chair would affect the task performance. Therefore, the laser was directly introduced to the microscope head via scanning mirrors consisting of a resonant mirror and a Galvano mirror. The laser power under the objective was 20–50 mW. Emitted fluorescence was split from the excitation light pathway by a dichroic mirror (reflection and transmission wavelength ranges of 400–755 nm and 800–1300 nm, respectively; NDM760, Olympus), and was collected with a cooled high-sensitivity GaAsP detector (Olympus) through a liquid light guide connecting the microscope head with the photodetector device. The fluorescence was then band-pass filtered (wavelength range, 400–760 nm; 32BA750 RIF, Olympus) and collected by a photomultiplier tube detector (PMT). The full-widths at half maximum of 2 μm fluorescent beads (Fluoresbrite YG Plain Microspheres; Polysciences, PA, USA) obtained by the two-photon imaging through two glass coverslips with the 10× objective were 1.29 ± 0.03 μm for the X-axis and 9.89 ± 0.17 μm for the Z-axis ($n = 5$ beads). The x and z resolutions of the beads imaged with the 25× objective were 1.11 ± 0.04 μm and 7.78 ± 0.15 μm ($n = 5$ beads), respectively. These values are comparable to those obtained in our previous study of two-photon calcium imaging of neuronal somata, dendrites, and axonal boutons in anesthetized marmosets[21]. The optical axis was adjusted to be nearly perpendicular to the plane of the cranial window by tilting the microscope body (5–20°). To shield the microscope objective from possible stray light, an aluminum foil dish was attached to the implanted metal chamber using a silicone elastomer (Kwik Cast, World Precision Instruments), and the space over the animal's head was covered with lightproof cloth. A series of 5000 images were acquired 1–5 times for the two-target reaching task, and 5–9 times for the force-field adaptation task, at a frame rate of 30 Hz and using FV30S-SW software (Olympus). The total imaging duration was 2.8–25.2 min (5000 frames corresponded to 2.8 min). When subcellular activity was imaged, the pixel size of the imaging field should be smaller than that used for imaging neuronal somata; the imaging fields were therefore set to 159 × 159 μm for dendrites and 85 × 85 μm for axons. The excitation light entering the cortex partially permeated through the eyes, allowing the eye movement to be tracked with the CCD camera. In some imaging experiments, the position of the left eye was recorded at 50 Hz and quantified using the "Analyze Particles" plugin in ImageJ (National Institute of Health, MD, USA), with default parameters.

**Image processing**. Time-lapse images were first realigned with a finite Fourier transform algorithm to remove tangential drifts[34]. The SDs of x and y shifts between the raw image frames and the motion-corrected image frames during the task performance were 1.06 ± 0.08 μm and 1.47 ± 0.13 μm, respectively ($n = 17$ imaging series from six imaging sessions from marmoset A), and 0.90 ± 0.05 μm and 0.48 ± 0.02 μm, respectively ($n = 23$ imaging series from six imaging sessions from marmoset D). The SDs of x and y shifts during periods that started 1.0 s before the pole movement and ended 1.0 s after the pole movement were 0.96 ± 0.05 μm and 1.17 ± 0.06 μm, respectively ($n = 337$ trials from six imaging sessions from marmoset A), and 0.43 ± 0.02 μm and 0.30 ± 0.02 μm, respectively ($n = 379$ trials from six imaging sessions from marmoset D).

Regions of interest (ROIs) corresponding to active neuronal somata (active ROIs) were extracted from the time series of the images using a constrained non-negative matrix factorization (CNMF) algorithm (http://github.com/epnev/ca_source_extraction; v0.42)[36]. The ROI number for the search was set at 100 for each field. Extracted ROIs with non-soma like contours and/or with only apparent noise were removed by visual inspection. For the dendrite and axon imaging data, ROIs were determined manually with ImageJ. For Figs. 7b and 9, the detrended relative change in fluorescence at a time point $t$ for each ROI was calculated as $\Delta F/F(t) = \frac{F(t) - F_0(t)}{F_0(t)}$, where $F(t)$ was the mean of the fluorescence intensity values of the pixels within the ROI at $t$ and $F_0(t)$ was calculated as the 8th percentile of the $F$ value across $t \pm 15$ s (corresponding to 900 frames). To calculate the full length of $F_0(t)$, traces were extended 450 frames with the value at the first frame before the first frame and 450 frames with the value at the last frame after the last frame. For the other image processing, denoised $\Delta F/F$ signals were used, with these signals being computed using the extract_DF_F function in the same CNMF package, with the parameters df_prctile set to 8 and df_window set to 900 frames. To identify ROIs from the same neurons over two sessions, the shifts between the imaging fields were corrected by the finite Fourier transform and NoRMCorre algorithms (https://github.com/flatironinstitute/NoRMCorre). The shifts were applied to the X-Y coordinates of the ROIs. The ROIs in the two imaging fields were then registered using the "registerROI" function in the CNMF package.

To analyze the activity in individual neurons, task-relevant neurons were first defined according to whether the mean denoised $\Delta F/F$ signals during the −1 to +2 s from the movement onset of the successful trials were significantly larger than those while the cursor stayed within the fixation square during the fixation period. Significance was assessed using the Wilcoxon rank-sum test ($P < 0.05$). For each task-relevant neuron in the two-target reaching task, the mean denoised $\Delta F/F$ signals during −1 to +2 s from the onset of the successful movement were averaged over target 1 reaching trials ($R1$) and target 2 reaching trials ($R2$). Then, the direction selectivity index (DSI) was defined as $\frac{R1-R2}{R1+R2}$. Thus, a DSI of $+1.0$ indicates that the neuron was active only during reaching to target 1. The task-relevant neurons in the force-field adaptation task were classified as follows: for each trial in the first baseline, first FF, and first washout blocks, the mean denoised $\Delta F/F$ during the period of −1 to +2 s from the movement onset was calculated. The trial series of the mean denoised $\Delta F/F$ in each block were then compared between the three blocks by using the Kruskal-Wallis test. Task-relevant neurons with a $P$-value of less than 0.05 were further classified into seven groups according to the difference between each pair of the blocks by using post-hoc test (Dunn-Sidak correction). The fractions of the seven groups of task-related neurons with a difference in activity between pairs of blocks were statistically tested as follows: for each neuron, the denoised $\Delta F/F$ traces were shuffled across successful trials and the same classification was performed with the new block labeling, allowing the fractions of these groups to be calculated. This was repeated 10000 times, and the 95th percentile values were determined.

To estimate a property of the activity in active ROIs, we calculated skewness, defined as the third central moment normalized to the cube of the standard deviation, of $\Delta F/F$ because it is an easily measurable indicator that can be used to pick up active ROIs with a relatively stable baseline and transient positive fluorescence changes at a biologically relevant frequency[35,68]. For each active ROI, the skewness of $\Delta F/F$ was calculated from the motion-corrected images and the mean value was 2.68 ± 0.09 ($n = 581$ active neurons in 14 fields from two marmosets). In our previous study, in which cortical neuronal activity was imaged in an anesthetized marmoset[21], 445 ROIs were manually determined and 81 ROIs with >0.5 skewness of $\Delta F/F$ were defined as active neurons. The skewness of $\Delta F/F$ in these active neurons was 2.23 ± 0.18 ($n = 81$ in three fields from one marmoset). When the CNMF algorithm was applied to the imaging data and the total ROI number for search was set to 450, 87 active ROIs were extracted and the skewness was 2.56 ± 0.17, which was comparable to that of manually detected active ROIs (skewness, $P = 0.13$, Wilcoxon's rank-sum test). This suggests that the automatically-extracted ROIs had similar skewness values to those of the manually detected ones.

**Statistics**. Statistics were performed using MATLAB (R2016a, 9.0.0.341360; MathWorks) or $R$ (3.1.2). Wilcoxon's signed-rank test, Wilcoxon's rank-sum test, Spearman's rank correlation coefficient test, Kruskal-Wallis test followed by post-hoc Dunn-Sidak test, and a random permutation test were used for statistical comparisons. Correlation coefficients were calculated from Spearman's correlations unless otherwise noted. No statistical tests were run to predetermine the sample size. Data are presented as mean ± SEM, unless otherwise noted. Blinding and randomization were not performed.

**Data availability**. The data supporting the findings of this study are available from the corresponding authors upon reasonable request.

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

## Acknowledgements

We thank E. Imoto, Y. Takahashi, M. Kondo, E. Iwase, Y. Takeda, and K. Kotani for animal handling. We thank V. Jayaraman, R. Kerr, D. Kim, L. Looger, and K. Svoboda of the GENIE Project Janelia Farm Research Campus (HHMI) for providing rAAV2/1-SynI-GCaMP6f. We thank K. Arai, Y. Saito, and A. Tsuchiya (Olympus) for helping to develop the two-photon microscopy. This work was supported by Grants-in-Aid for Scientific Research on Innovative Areas (17H06025 and 17H06309 to M.M., 16H01492 and 18H05059 to Y.M., 16H01624 to Y.R.T., 15H05873 to A.N.), for Scientific Research (A) (15H02350 to M.M., 26250009 to A.N.), for Young Scientists (A) (17H04982 to Y. M.), and for Young Scientists (B) (17K13277 to T.E., 16K18370 to Y.R.T.) from the Ministry of Education, Culture, Sports, Science, and Technology, Japan; AMED (JP17dm0207050 and JP16dm0207046 to A.N., JP17dm0107051 to E.S., JP17dm0207001 to T.Y., and JP17dm0107053, JP17dm0207027, and JP17dm0107150 to M.M.); the Nakajima Foundation (to Y.M.); the Konica Minolta Science and Technology Foundation (to Y.R.T.); and the Tokyo Society of Medical Sciences (to Y.M.).

## Author contributions

T.E., Y.M., and M.M. designed the experiments. T.E., Y.M., Reiko H., and Y.H. conducted the experiments. T.E., Y.M., Riichiro H., D.K., A.N., K.H., and E.S. designed the marmoset chair and head holder, and T.E., Y.M., Reiko H., and Y.H. optimized the task apparatus for the experiments. T.E., Y.M., Riichiro H., S-I.T., D.K., and A.N. performed intracortical microstimulation, and A.W., H.M., and T.Y. designed and prepared an AAV vector system. T.E. and Y.M. analyzed behavioral data, and T.E., Y.M., and Y.R.T. analyzed imaging data. T.E., Y.M., and M.M. wrote the paper, with comments from all the authors.

## Additional information

**Competing interests:** The authors declare no competing interests.

