## [Peer Review File · Nature Communications]

Reviewers' comments:

Reviewer #1 (Remarks to the Author):

Ebina and colleagues investigated neural dynamics in the motor cortex of the common marmoset during upper-limb movements. They trained four marmosets on different tasks (visually cued pole-pull task, target reaching task and force-field adaption task) and used two-photon calcium imaging to record signals from layer 3 neurons in two animals. They demonstrate that body-restrained marmosets learn internal- and external stimulus-triggered reaching tasks which are commonly used in other NHPs studies. They successfully use two-photon imaging in awake, behaving marmosets and follow the activity of neurons in different sessions over several days.

The authors give a broad overview on the training progress and learning possibilities of head-fixed common marmosets. This animal model possesses several advantages for electrophysiological as well as imaging studies and the authors show here that it is possible to train the animals on motor-skill tasks. The adjustments they made to the restraining apparatus and the two-photon microscope are convincing. Overall, I found this study interesting and the methods are impressive, but the paper was largely methodological and might be more appropriate as a methods-type paper.

Major comments

1. The authors emphasize that ensemble representations are stable whereas single neurons are not. It is unclear in what sense ensemble representations are stable. The only thing that is relatively stable is the decoding performance. But the decoder has to be re-calibrated on every session so the decoder is clearly not stable. I would recommend that the authors downplay the point that ensemble representations are stable.
2. Methods: For the different tasks one animal (two-target reaching task) and up to three animals (visually cued pole-pull task) were trained. Marmosets are known to be very sensitive to re-restraining procedures (see Yamada et al., 2016, Scientific Reports). Did the authors use just the four animals for the acclimation (which would mean a 100% success rate) or did they start with more animals for the acclimation to the apparatus and report results from the four animals that successfully acclimatized?
3. Imaging data: The numerical aperture of 0.6 and 10x objective for imaging the neuronal soma-ta seems to be very low. What are the reasons for choosing these parameters? Additionally, it is difficult to identify the cell somata in Figure 6. The neuronal somata of neuron 2 in Figure 6 cannot be spotted clearly. If the somata are clearly distinguishable, a magnification and marking of the area which was used for the analysis should be shown.

ImageJ provides a plugin that automatically detects region of interests. Why did the authors decided to do it manually?

4. Does the population of neurons (Figure 6 and lines 262f) always include the same 32 neurons in each session? How were these neurons tracked across and throughout sessions? Did the authors try the prediction with different population sizes and "qualities" of neurons?
5. How many neurons can be grouped into the distinct characteristics (described in Figure 7) in each session? And how many of those can be identified across sessions?

Minor comments

1. Lines 295f: In how many sessions/ trials? Was it possible to image the dendrites during the different task blocks?
2. It is maybe advisable to put task- and animal-specific parameters which are important for the tasks in a table (including which animal was trained for which task, how many sessions of imaging...).

3. What was the time line of surgery and virus injection relative to the training? (Line 232 f).
4. Line 129f: Are these results significant?
5. Figure 4: Please check 4c. The baseline of the early 10 trials are missing and force field data for early 10 days seem to be cut off. It would be nice to include the regression line in the magnification of the washout data (4d).

Reviewer #2 (Remarks to the Author):

In this study, the authors reported their findings from a study in which they were able to train marmosets in several motor tasks and then image neural activity of motor cortex while marmosets were performing the tasks. This is one of the first studies that applies two-photon imaging techniques in marmosets during behavioral tasks. The observations reported in this manuscript are of broad interest to this field. This manuscript contains two parts, a detailed description of motor behavior training procedures and a fairly superficial description of two-photon imaging data acquired during the task. The strength of this manuscript is the behavioral methodologies developed by these authors. The conclusions drawn from the imaging data, however, were not fully supported by the evidence presented.

Specific comments:

1. Much of this manuscript is devoted to the descriptions of procedures developed to train marmosets to perform motor tasks (Figures 1-4). The authors developed innovative and creative methods (apparatuses and paradigms) to train marmosets in motor tasks. Because marmosets are relatively new to behavioral neuroscience and only a limited numbers of groups have attempted to train these animals (comparing to macaque monkeys), there has been a lot of skeptics on what tasks can be learnt by marmosets. This study showed very nicely that if you designed apparatuses and paradigms that are appropriate for marmosets (instead of simply copying them from macaque monkeys), you should be able to train marmosets in a wide range of tasks. This part of the study is very creative and nicely done. My only comment is that the authors should consider comparing some aspects of marmosets' motor performance to those by macaque monkeys or humans (data from other published studies) wherever appropriate. Such comparisons make the marmoset data more interesting to a broad range of readers. For example, Mitchell et al. (J Neurosci 2014) compared marmoset's visual perception (saccade) with those of macaque monkeys, Song et al. (PNAS, 2016) compare marmoset's auditory perception (pitch) data with those of humans.

2. The second part of the manuscript (Figures 5-8) describes two-photon imaging data. The methods to conduct the two-photon imaging are impressive, but the functional conclusions provided by the authors are on a shaky ground, based on rather limited data, small sample sizes (see more comments below). I suggest the authors to tune down these conclusions. What these data showed was the feasibility of conducting two-photon imaging measurements in marmosets performing motor tasks, which is significant by itself.

3. Line 232: How far apart were injections?

4. Line 258: Were "32 neurons" from one marmoset?

5. Lines 279-281: It appears that this statement ("These results suggest that layer-3 neurons ... ") is based on data from only one marmoset (shown in Figure 6). This is a very thin evidence for a rather big conclusion. The authors should at least repeat these observations in multiple marmosets.

6. Lines 291-293: This statement is not really informative. Again, it is based on very little data shown on Figure 7.

Reviewer #3 (Remarks to the Author):

Ebina et al. show in their study "Two-photon imaging of neural dynamics in the motor cortex of non-human primates during upper-limb movement tasks" that head-fixed common marmosets can be used for motor-related behavior tasks in combination with two-photon calcium imaging. They trained four marmosets to perform various reaching tasks. In a next step, GCaMP6-expressing neurons in the motor cortex were imaged in various sessions over the course of multiple days. The authors describe a promising combination of tools for future brain research using non-human primates. However, as the authors themselves note in their introduction, earlier studies have reported already such behavioral as well as two-photon imaging experiments in the awake marmoset before. For example, the restraining system to cope with body movements during upper-limb behavior is heavily based on a previously published system (Schultz-Darken et al., 2004). Nevertheless, a useful aspect of the study is the development of improved protocols and hardware, which integrate existing marmoset-handling and -imaging techniques. Instead, the new insights on upper limb motor control by the cortex are very limited and at best at a proof-of-concept stage. In conclusion, despite the limited novelty but in view of the increasing general interest in marmoset cellular in vivo studies, the manuscript might be suitable for consideration as a resource/methods paper.

Major points

1. The authors need to include a clear step-by-step protocol of their method, with the timelines of the individual training steps, a list of caveats for the relevant steps of the training and more detailed schematics about the experimental setup. The photographs in figure 1 and 5 give a general idea about the hardware, but specific dimensions and design instructions are needed.
2. More details on two-photon calcium imaging are needed. What was the extent of variability in imaging quality (signal-to-noise ratio at similar recording depths) between different experiments? What are the critical parameters that need to be optimized? How long is the training needed to reduce unwanted body movements during imaging in awake animals? How many times can the same cells/dendrites be imaged?

Minor points

Line 108: Please describe the procedure in more detail, providing the presumed changed session duration over time.

Line 473: More details of the visual stimulation setup itself are needed, including the distance of screen from eye, screen intensity etc.

Figure 2: What is the reason for the signal fluctuations in panel g: between sessions 14-17, there is a consistent drop of the number of hits per trial, which is unlikely to be just noise.

Line 600 onward: The image processing paragraph is difficult to understand without accessing other manuscripts. Please elaborate more on "five-frame averaging", e.g. does this describe a moving-average filter etc.? How was dF/F calculated at the beginning of the recording, if it is based on ± 15 s around each sample time point? In line 601-602, what "skewness" of which distribution?

Reviewer #4 (Remarks to the Author):

This interesting paper combines two fundamental advances, controlled reaching tasks in marmosets, and two-photon calcium imaging in a primate performing a complex task. As such, the work is of great interest for the field. My only major criticism of the paper is that the analysis of the two-photon data is very limited. In particular, the current analysis does not take advantage of the benefits of two-photon imaging (chronic imaging of the same neurons, and imaging from large groups of neurons with known spatial relationships). The paper could be significantly improved by expanding this part.

Major comments:

- With the exception of a few correlation analyses, there are no statistics for any of the behavioral tasks (are training effects significant, for example?).
- The two-photon analysis should be significantly expanded. Some possible venues for additional analyses are described below.
 - o Target reaching task: What are the basic tuning properties of these neurons? Do they respond more for one target direction than the other? Does the tuning change with training? What about activity in error trials? How are responses across neighboring neurons correlated, both during successful and non-successful trials? Do these spatial correlation patterns change with training? Do certain neurons reliably respond during certain parts of the movement trajectory?
 - o Force-field adaptation: Similar to above – what are the basic tuning properties and spatial correlation patterns? Do neurons that change or fail to change their responses in the different blocks cluster?
 - o Dendrites: How do different parts of the same dendritic tree differ in their tuning properties? Are boutons clustered according to their response properties? Can the authors follow dendrites to their respective somata and compare tuning properties of dendrites with those of somata?

Minor comments:

- The introduction is rather unfocused, which makes some of the statements misleading (for example, depending on the brain area neurons can stably represent single parameters; two-photon imaging has been used for a lot more than just population dynamics in working memory etc). In addition, the emphasis on population dynamics in both abstract and introduction seems inappropriate, as these are not further discussed or analyzed in the paper. It would be better to limit the scope of the introduction to movement tasks and the promise of two-photon imaging in their context.
- It would be helpful to clarify earlier that the initial tasks are not performed with head fixation.
- Visually cued pull task:
 - o Is the threshold across which animals have to pull the pole the same across animals?
 - o Do the animals have to hold on to the pole in the blank period?
 - o In addition, some discussion of the difference between animals seems warranted.
- Target reaching task:
 - o The description of the orthogonality of the two axes in the main text should be improved (it is not immediately clear that leftward is orthogonal to a movement direction toward the animal).
 - o Similarly, the main text should explain that the target is straight below the fixation square, otherwise the rest of the analysis is hard to follow.
 - o Why is there a force field applied during the blank period only?
 - o The correlation analysis doesn't seem appropriate for these data, which show changes in only very few, early sessions.
 - o There should be some discussion regarding the fact that while the trajectories in both monkeys indeed became straighter, one of the animals failed to perform much above 50% for the majority of the sessions. What happens in these error trials?

- Force-field adaptation:

o The authors state that the force-field remained on for 20 consecutive trials. How many trials total does that constitute? Is there a different failure rate in the early versus late block?

We thank all reviewers for their careful consideration of our manuscript and for making helpful comments. Our detailed responses to the reviewers' comments are provided below:

Reviewer #1: Ebina and colleagues investigated neural dynamics in the motor cortex of the common marmoset during upper-limb movements. They trained four marmosets on different tasks (visually cued pole-pull task, target reaching task and force-field adaption task) and used two-photon calcium imaging to record signals from layer 3 neurons in two animals. They demonstrate that body-restrained marmosets learn internal- and external stimulus-triggered reaching tasks which are commonly used in other NHPs studies. They successfully use two-photon imaging in awake, behaving marmosets and follow the activity of neurons in different sessions over several days.

The authors give a broad overview on the training progress and learning possibilities of head-fixed common marmosets. This animal model possesses several advantages for electrophysiological as well as imaging studies and the authors show here that it is possible to train the animals on motor-skill tasks. The adjustments they made to the restraining apparatus and the two-photon microscope are convincing. Overall, I found this study interesting and the methods are impressive, but the paper was largely methodological and might be more appropriate as a methods-type paper.

Major comments

1. The authors emphasize that ensemble representations are stable whereas single neurons are not. It is unclear in what sense ensemble representations are stable. The only thing that is relatively stable is the decoding performance. But the decoder has to be re-calibrated on every session so the decoder is clearer not stable. I would recommend that the authors downplay the point that ensemble representations are stable.

In the revised manuscript, we have re-analyzed the imaging data with a fully automated ROI detection and registration algorithm, as described in the responses to the third and fourth comments. This approach tracked only six neurons over all of the six sessions. This low number is insufficient to analyze the ensemble representation. Therefore, we have removed the description of the stability of the ensemble representation across the sessions.

Instead, in the revised manuscript, we have added data on multi-day imaging during the two-target reaching task using another marmoset (marmoset D). Then, we used the imaging data from the two marmosets to analyze the target-selectivity of the activity of individual neurons across the sessions. We found that some neurons exhibited the same selectivity during the multi-day imaging sessions, and that the distribution of the target-selectivity was slightly biased towards the target-1 direction (corresponding to pulling of the manipulandum) in both marmosets. We have added these results to Fig. 7.

2. Methods: For the different tasks one animal (two-target reaching task) and up to three animals (visually cued pole-pull task) were trained. Marmosets are known to be very sensitive to restraining procedures (see Yamada et al., 2016, Scientific Reports). Did the authors use just the four animals for the acclimation (which would mean a 100% success rate) or did they start with more animals for the acclimation to the apparatus and report results from the four animals that successfully acclimatized?

We trained just four marmosets (marmosets A, B, C, and D) in this study, and all of them were successfully acclimated to the apparatus and head fixation, and performed the given tasks. Before starting the training with a jacket body restraint, we tried to acclimatize another marmoset to a chair modified from a traditional macaque chair, but failed. Even when the head movement was restricted by a neck plate, but the marmoset could still rotate their head on the chair, the marmoset did not acclimatize to pulling the pole. Therefore, in this study we developed new step-by-step training protocols with the body restrained by a jacket, which started from the habituation step without head fixation.

3. Imaging data: The numerical aperture of 0.6 and 10x objective for imaging the neuronal soma-ta seems to be very low. What are the reasons for choosing these parameters?

In the marmoset, the skull is thicker than in the mouse, and the distance between the skull and dura mater is longer. Therefore, the distance between the objective exit and the cortical surface should be longer than when imaging the mouse neocortex. This is the reason why we used an objective with a long working distance (8 mm). In our

previous study (Sadakane et al., Cell Rep. 13, 1989-1999, 2015), we estimated the spatial resolution using the same objective and a cranial window (lateral resolution of 2 μm beads, $\sim 2.0 \mu\text{m}$, axial resolution of 2 μm beads, $\sim 8.5 \mu\text{m}$), and demonstrated that this objective can be used for two-photon calcium imaging of neuronal somata, dendrites, and axonal boutons in the somatosensory cortex of anesthetized marmosets.

Additionally, it is difficult to identify the cell somata in Figure 6. The neuronal somata of neuron 2 in Figure 6 cannot be spotted clearly. If the somata are clearly distinguishable, a magnification and mark-ing of the area which was used for the analysis should be shown.

In the revised manuscript we have extracted all ROIs corresponding to neuronal somata using fully automated ROI detection and registration algorithms described in the next response. We have also added magnified images of representative neuronal somata and the corresponding ROI contours to Figure 7a.

ImageJ provides a plugin that automatically detects region of interests. Why did the authors decided to do it manually?

In the previous version, we manually detected ROIs, as this was the method we used in our previous study of two-photon calcium imaging of cortical neurons in anesthetized marmosets (Sadakane et al., Cell Rep. 13, 1989-1999, 2015). According to the reviewer's comment, we have now fully automated the detection and registration of ROIs by combining CNMF (http://github.com/epnev/ca_source_extraction; Pnevmatikakis et al., Neuron 89, 285-299, 2016) and NoRMCorre (<https://github.com/flatironinstitute/NoRMCorre>) programs, and we have re-analyzed the activity of individual neurons. We have described the details of the ROI detection and registration in the Methods section (lines 720–736).

4. Does the population of neurons (Figure 6 and lines 262f) always include the same 32 neurons in each session? How were these neurons tracked across and throughout sessions? Did the authors try the prediction with different population sizes and “qualities” of neurons?

In the previous manuscript, we registered the same ROIs across multiple imaging sessions by manual visual inspection, according to their relative locations in the field,

and pursued the 32 neurons over all of the six sessions. In the revised manuscript, we adopted the CNMF algorithm to detect ROIs in each field as described above, and then used NoRMCorre and registerROI functions in the CNMF package to track the ROIs of the same neurons across the sessions. As a result, the number of neurons detected in every imaging session in the same marmoset decreased from 32 to 6. This reduction might be because the CNMF algorithm does not extract neurons with a very low frequency of calcium transients, even when the fluorescent structure is detected in the same location by visual inspection. This number of six was too small for the population analysis. We have therefore omitted the population analysis from the revised manuscript.

Regarding the qualities of the neurons, we used the constrained_foopsi script in the CNMF package to estimate the median of the standard deviation of the high-frequency components of the fluorescence signals in each pixel of the images. This value is thought to reflect the motion artifact and signal-to-noise ratio in the fluorescence signals. It did not change across imaging sessions, indicating that the difference in the imaging quality did not apparently affect the difference in the neuronal activity between sessions. We have added this result in Supplementary Fig. 5.

5. How many neurons can be grouped into the distinct characteristics (described in Figure 7) in each session?

We increased the imaging data during the adaptation task using another marmoset (marmoset A), and analyzed these additional data. From seven imaging fields in marmoset A, 113 active neurons were extracted, and 56 neurons were defined as task-relevant. From eight imaging fields in marmoset D, 425 active neurons were extracted, and 163 neurons were defined as task-relevant. The task-relevant neurons were defined as those neurons with significantly higher $\Delta F/F$ signals during the period of -1 to $+2$ s from the movement onset in comparison with the fixation period. According to the amplitude of the task-relevant activity averaged over successful trials in each block, we classified the task-relevant neurons into four types for each marmoset: “FF-changed”, “BW-changed”, “unchanged” and “other” neurons. FF-changed neurons were defined as task-relevant neurons that exhibited significant differences in the mean task-relevant activity between baseline and FF blocks and between FF and washout blocks, but not between baseline and washout blocks. BW-changed neurons were defined as task-relevant neurons that exhibited significant differences in the mean

task-relevant activity between baseline and washout blocks. Unchanged neurons were defined as task-relevant neurons that did not exhibit any significant difference between pairs of the three blocks. Other neurons were defined as the other task-relevant neurons. As the mean cursor trajectory in the FF block deviated substantially from those in the baseline and washout blocks, and the mean cursor trajectory in the baseline block was similar to that in the washout block, we hypothesized that FF-changed neurons might represent the cursor trajectory, and BW-changed neurons might be a subset of neurons that changed their activity as the block progressed. The FF-changed neurons numbered 8 (7 fields from marmoset A) and 17 (8 fields from marmoset D). The BW-changed neurons numbered 28 (7 fields from marmoset A) and 83 (8 fields from marmoset D). The fractions of these neurons to the task-relevant neurons were significant in comparison with those obtained from shuffled data (for all cases, $P < 0.05$). We have added these results to Figure 8.

And how many of those can be identified across sessions?

In the force-field adaptation task, we imaged different fields in different sessions. Therefore, we could not pursue the same neurons across different sessions.

Minor comments

1. Lines 295f: In how many sessions/ trials? Was it possible to image the dendrites during the different task blocks?

In this study, we attempted to detect the dendritic activity in 15 fields. In 9 of the 15 imaging fields we detected movement-related activity of dendrites (dendrite-like structures extracted from the map of correlation coefficients to the cursor trajectory, as shown in Fig. 9c). We have described the experimental result in lines 356–363. As we performed this experiment only during the nFF blocks of the adaptation task, we could not compare the response differences of the dendrites between the task blocks. Although it might be possible to image the dendrites during the different task blocks, we think that many more imaging experiments would be required to characterize the task-relevant activity, and such experiments are beyond this study. Thus, we focused on demonstrating the feasibility of two-photon calcium imaging of neuronal somata.

2. It is maybe advisable to put task- and animal-specific parameters which are

important for the tasks in a table (including which animal was trained for which task, how many sessions of imaging...).

3. What was the time line of surgery and virus injection relative to the training? (Line 232 f).

We have included the time-line of the task training and the timing of the virus injection, attachment of the head plate, and two-photon imaging for each marmoset in Supplementary Fig. 1. The number of training sessions for each task and each marmoset has also been included in Supplementary Fig. 1.

4. Line 129f: Are these results significant?

Yes, these results are significant. The increase of the number of hits per cue period was statistically significant for all of the three marmosets (evaluated by the Spearman correlation coefficients between the numbers of hits and the session; 0.81 for marmoset A, 0.71 for marmoset B, and 0.81 for marmoset C; $P < 0.01$ in all marmosets). We have added these values to the legend of Figure 2. In addition, we have added the statistical results for Fig. 2e–h to the corresponding legends.

5. Figure 4: Please check 4c. The baseline of the early 10 trials are missing and force field data for early 10 days seem to be cut off. It would be nice to include the regression line in the magnification of the washout data (4d).

We thank you for this comment. In the previous manuscript, these example trajectories were derived from the second nFF block (baseline), second FF block (force-field), and third nFF block (washout) within the same session. Thus, in these trajectories, the early 10 trials in the baseline block corresponded to the early 10 trials with slight after-effects from the preceding washout block. We were afraid that it might be confusing and did not show these trials. In the revised manuscript, we have displayed all trajectories in the early and late 10 trials from the first nFF block (the first baseline), first FF block (the first force-field), and second nFF block (the first washout) within the same session in Figure 5c. We have added a regression line to the magnification of the washout data in Figure 5d.

Reviewer #2: In this study, the authors reported their findings from a study in which they were able to train marmosets in several motor tasks and then image neural activity of motor cortex while marmosets were performing the tasks. This is one of the first studies that applies two-photon imaging techniques in marmosets during behavioral tasks. The observations reported in this manuscript are of broad interest to this field. This manuscript contains two parts, a detailed description of motor behavior training procedures and a fairly superficial description of two-photon imaging data acquired during the task. The strength of this manuscript is the behavioral methodologies developed by these authors. The conclusions drawn from the imaging data, however, were not fully supported by the evidence presented.

Specific comments:

1. Much of this manuscript is devoted to the descriptions of procedures developed to train marmosets to perform motor tasks (Figures 1-4). The authors developed innovative and creative methods (apparatuses and paradigms) to train marmosets in motor tasks. Because marmosets are relatively new to behavioral neuroscience and only a limited numbers of groups have attempted to train these animals (comparing to macaque monkeys), there has been a lot of skeptics on what tasks can be learnt by marmosets. This study showed very nicely that if you designed apparatuses and paradigms that are appropriate for marmosets (instead of simply copying them from macaque monkeys), you should be able to train marmosets in a wide range of tasks. This part of the study is very creative and nicely done. My only comment is that the authors should consider comparing some aspects of marmosets' motor performance to those by macaque monkeys or humans (data from other published studies) wherever appropriate. Such comparisons make the marmoset data more interesting to a broad range of readers. For example, Mitchell et al. (J Neurosci 2014) compared marmoset's visual perception (saccade) with those of macaque monkeys, Song et al. (PNAS, 2016) compare marmoset's auditory perception (pitch) data with those of humans.

We thank you for this comment. The marmosets' performance with the manipulandum is comparable with that of macaques, because the cursor trajectory straightened after one–two weeks. The duration of the training was comparable to that used in macaques. However, the one- or two-target reaching tasks were easier than an eight-direction

cursor movement task performed by macaques. The marmosets could perform the same task for more than one month without losing motivation. Thus, as in the macaque electrophysiological experiments, the experimenters could acquire much imaging data from the behaving marmosets. However, some of the motor skills described in this study should be improved. One problem is the smaller number of trials in each training session (100–200) in comparison with the numbers used for macaques (> 480 ; Li et al., *Neuron* 30, 593–607, 2001). In the adaptation task, we set 20 successful trials in each block, which may have prevented the return of the cursor movement to the baseline trajectory during the perturbation block, as was observed in macaques and humans. This could possibly be overcome by reducing the reward for each trial. An increased number of trials in each session should significantly improve the performances, and further shorten the training sessions for a task. Then, if the two-target reaching task could be extended to an 8-direction reaching task, it could be said that the marmoset can perform similar motor tasks to macaques and humans. In the revised version of the manuscript, we discussed these points in the Discussion section (lines 376–391).

2. The second part of the manuscript (Figures 5-8) describes two-photon imaging data. The methods to conduct the two-photon imaging are impressive, but the functional conclusions provided by the authors are on a shaky ground, based on rather limited data, small sample sizes (see more comments below). I suggest the authors to tune down these conclusions. What these data showed was the feasibility of conducting two-photon imaging measurements in marmosets performing motor tasks, which is significant by itself.

We agree with the reviewer's comment that our previous conclusions were based on limited data. We have tuned-down the functional conclusions based on the task-relevant neuronal activity. Instead, to more firmly demonstrate the feasibility of conducting two-photon imaging in a marmoset performing motor tasks, we first trained another marmoset on both the two-target reaching task and the adaptation task, to increase the imaging data. Second, we demonstrated that many ROIs surrounding neurons can be extracted using fully automated ROI detection and registration algorithms, as have been used in mouse studies. Third, we calculated the target selectivity of the activity of individual neurons, as is usually performed in the macaque, and estimated its stability over days. We also classified the activity of individual neurons in the adaptation task into four groups according to the activity differences between blocks with and without a force field, modifying the classifications used in macaque electrophysiology studies

(Ganfold et al., PNAS, 97, 2259-2263, 2000; Li et al., Neuron 30, 593-607, 2001). We found similar results in two marmosets, which strengthened the feasibility of the calcium imaging and analysis of the neuronal activity in marmosets performing upper-limb-movement tasks. We have added these results to Figures 7 and 8.

3. Line 232: How far apart were injections?

The injection sites were 0.5–1 mm apart from each other. We have added this sentence to line 482.

4. Line 258: Were “32 neurons” from one marmoset?

Yes, the 32 neurons were from the same imaging field in marmoset A, and these were identified manually. However, in the revised manuscript we adopted the fully automated ROI detection and registration methods described above, and analyzed a different set of ROIs to those used in the previous dataset. The number of neurons that were imaged in every imaging session decreased from 32 to 6 in marmoset A. This number is insufficient for analysis of the population dynamics, so we have removed this analysis from the revised manuscript.

5. Lines 279-281: It appears that this statement (“These results suggest that layer-3 neurons ...”) is based on data from only one marmoset (shown in Figure 6). This is a very thin evidence for a rather big conclusion. The authors should at least repeat these observations in multiple marmosets.

Following this comment, we have repeated the two-photon imaging in another marmoset (marmoset D) performing the two-target reaching task. In marmoset A, only six automatically extracted neurons were tracked across all six sessions. In marmoset D, it was only three. These numbers are insufficient for analysis of the population dynamics. Therefore, we have removed the analysis of the ensemble representation. Instead, as reviewer 4 pointed out, we have examined the target-selective activity of individual neurons. We defined the movement direction selectivity index (DSI) in successful trials for each neuron in the imaging field. In DSI, a value of 1 indicates that neurons exhibited calcium transients in only target 1-directed movement, but not in target 2-directed movement, and -1 indicates that neurons exhibited calcium transients in only target 2-directed movement, not in target 1-directed movement. We found that

49 and 22 out of 215 (the cumulative total number) active neurons from the two marmosets were selective to target 1- and target 2-movements respectively (neurons with $DSI > 0.5$ and < -0.5 respectively). The DSI was correlated between days for each marmoset, and its distribution was slightly biased to positive (that is, target 1-direction) in both marmosets. We have added these results to Figure 7 in the revised manuscript.

6. Lines 291-293: This statement is not really informative. Again, it is based on very little data shown on Figure 7.

Following this comment, we have repeated the two-photon imaging in another marmoset during the force-field adaptation task. From seven imaging fields in marmoset A, 113 active neurons were extracted and 56 neurons were defined as task-relevant. From eight imaging fields in marmoset D, 425 active neurons were extracted and 163 neurons were defined as task-relevant. The task-relevant neurons were defined as those neurons with significantly higher $\Delta F/F$ signals during the period of -1 to $+2$ s from the movement onset in comparison with the fixation period. According to the amplitude of the task-relevant activity averaged over successful trials in each block, we classified these neurons into four types for each marmoset: “FF-changed”, “BW-changed”, “unchanged” and “other” neurons. FF-changed neurons were defined as task-relevant neurons that exhibited significant differences in the mean task-relevant activity between baseline and FF blocks and between FF and washout blocks, but not between baseline and washout blocks. BW-changed neurons were defined as task-relevant neurons that exhibited significant differences in the mean task-relevant activity between baseline and washout blocks. Unchanged neurons were defined as task-relevant neurons that did not exhibit any significant difference between pairs of the three blocks. Other neurons were defined as the other task-relevant neurons. As the mean cursor trajectory in the FF block substantially deviated from those in the baseline and washout blocks, and the mean cursor trajectory in the baseline block was similar to that in the washout block, we hypothesized that FF-changed neurons might represent the cursor trajectory and BW-changed neurons might be a subset of neurons that changed their activity as the blocks progressed. The FF-changed neurons were 14.3% and 10.4% of the task-relevant neurons in marmosets A and D respectively. The fractions of the BW-changed neurons were 50.0% and 50.9% respectively. All these values were significant compared with those obtained from shuffled data ($P < 0.05$). We have added these results to Figure 8.

Reviewer #3: Ebina et al. show in their study “Two-photon imaging of neural dynamics in the motor cortex of non-human primates during upper-limb movement tasks” that head-fixed common marmosets can be used for motor-related behavior tasks in combination with two-photon calcium imaging. They trained four marmosets to perform various reaching tasks. In a next step, GCaMP6-expressing neurons in the motor cortex were imaged in various sessions over the course of multiple days. The authors describe a promising combination of tools for future brain research using non-human primates. However, as the authors themselves note in their introduction, earlier studies have reported already such behavioral as well as two-photon imaging experiments in the awake marmoset before. For example, the restraining system to cope with body movements during upper-limb behavior is heavily based on a previously published system (Schultz-Darken et al., 2004). Nevertheless, a useful aspect of the study is the development of improved protocols and hardware, which integrate existing marmoset-handling and -imaging techniques. Instead, the new insights on upper limb motor control by the cortex are very limited and at best at a proof-of-concept stage. In conclusion, despite the limited novelty but in view of the increasing general interest in marmoset cellular in vivo studies, the manuscript might be suitable for consideration as a resource/methods paper.

Major points

1. The authors need to include a clear step-by-step protocol of their method, with the timelines of the individual training steps, a list of caveats for the relevant steps of the training and more detailed schematics about the experimental setup. The photographs in figure 1 and 5 give a general idea about the hardware, but specific dimensions and design instructions are needed.

In line with these comments, we have added the training time-lines for each marmoset in Supplementary Fig. 1, and a list of caveats for each step in Table 1. We have clearly rewritten the step-by-step protocols of the methods with corresponding subtitles in the Methods section, to make it easier for readers to understand (lines 524–562). We have added projection views (with dimensions) of the marmoset chair (Supplementary Fig. 2) and the X-Y slide table to connect the pole and robotic arm for the 2D manipulandum (Supplementary Fig. 4). We have also added information on the characteristics of the dichroic mirror and filters used in the microscopy (Fig. 6). As the major microscopy

design belongs to the Olympus Corporation, its details are difficult to present in this study. However, this microscopy may be available if it is requested to Olympus.

2. More details on two-photon calcium imaging are needed. What was the extent of variability in imaging quality (signal-to-noise ratio at similar recording depths) between different experiments?

We measured the noise level of the imaging data observed during the two-target reaching and force-field adaptation tasks. The noise level was defined as the median value of the standard deviation of the high-frequency components of the fluorescence signals in each pixel of the images estimated by the `constrained_foopsi` script in the CNMF package (http://github.com/epnev/ca_source_extraction). This value is thought to reflect the motion artifact and signal-to-noise ratio in the fluorescence signals. The noise level was not correlated with the imaging depth (ranging from 120–325 μm during the force-field adaptation task), and the noise level (at a fixed depth of 250 μm during the two-target reaching task) did not change across the imaging sessions (Supplementary Fig. 5). Thanks to the reviewer's comment, we think the demonstration of the feasibility of acquiring multi-day two-photon calcium imaging of neuronal somata in the superficial layer of head-fixed marmosets performing motor tasks is now strengthened.

What are the critical parameters that need to be optimized? How long is the training needed to reduce unwanted body movements during imaging in awake animals?

From our experience, the critical point is to train marmosets in a step-by-step manner, as we have described in the manuscript. The most important point is the training with the body restraint jacket. Before starting this restraint, we tried to acclimatize another marmoset to a chair modified from a traditional macaque chair, but failed. Even when head movement was restricted by a neck plate, but the marmoset could still rotate its head on the chair, the marmoset did not become accustomed to the pole-pull behavior. Therefore, we developed new step-by-step training protocols with body restraint using the jacket, which started from the habituation step without head fixation. Our results demonstrated that marmosets can perform the pole-pull task with head-fixation within approximately 30 sessions. There was not a specific caveat for starting the imaging session. Therefore, we think that approximately 30 training sessions could be required

to reduce unwanted body movements during the imaging of awake animals (see Supplementary Figure 1 for the training schedule and Table 1 for caveats in each step of the tasks).

How many times can the same cells/dendrites be imaged

According to our new ROI detection and registration algorithm using CNMF and NoRMCorre (<https://github.com/flatironinstitute/NoRMCorre>) programs, we tracked 28 active neurons across 2 or more imaging sessions in marmoset A, and 53 active neurons in marmoset D. Of these, 6 and 3 active neurons were detected across all of the six sessions. Although we did not attempt to image the same field for another session, we think it would be possible, because we did not detect an increase in the noise level or apparent tissue damage during the imaging sessions, and the laser power under the objective was moderate (20–50 mW). We did not perform multi-day dendrite or axon imaging because we focused on demonstrating the feasibility of multi-day two-photon calcium imaging of neuronal somata in behaving marmosets.

Minor points

Line 108: Please describe the procedure in more detail, providing the presumed changed session duration over time.

We have added a detailed description of the training duration and the number of rewards during the training of the head-fixed pole-pull task. We have also added the mean values of these parameters during the training of the pole-pull task without head fixation. We have added these data as Supplementary Figure 3.

Line 473: More details of the visual stimulation setup itself are needed, including the distance of screen from eye, screen intensity etc.

We have added a description of the intensity of visual stimulation (cursor, fixation square, and target square or rectangle) and the distance of the screen from the eye to the Methods section (lines 568, 571, 595, 596, 637, 638, and 655).

Figure 2: What is the reason for the signal fluctuations in panel g: between sessions 14-17, there is a consistent drop of the number of hits per trial, which is unlikely to be just noise.

Between sessions 14 and 15, marmoset B was not trained for 11 days, because the marmoset had diarrhea. This long period without training might have caused the performance drop.

Line 600 onward: The image processing paragraph is difficult to understand without accessing other manuscripts. Please elaborate more on “five-frame averaging”, e.g. does this describe a moving-average filter etc.? How was $\Delta F/F$ calculated at the beginning of the recording, if it is based on ± 15 s around each sample time point? In line 601-602, what “skewness” of which distribution?

We are sorry that these descriptions were confusing. In the previous manuscript, the five-frame averaging referred to a moving-average filter having a window of ± 2 frames from the target frame. This means that the $\Delta F/F$ at the t -th frame was calculated as the mean of the $\Delta F/F$ signals during $t \pm 2$ frames. To calculate $\Delta F/F$ around the beginning and ending of the imaging, traces were extended 450 frames with the value at the first frame before the first frame and 450 frames with the value at the last frame after the last frame. These extended frames were used to calculate the all length of $F_0(t)$ for calculating $\Delta F/F$ around the beginning and ending of the imaging. The skewness was calculated from all of the $\Delta F/F$ signals during each imaging session for each cell. However, in the revised manuscript, as described above, we automatically extracted the ROIs and their denoised $\Delta F/F$ signals using the CNMF package for MATLAB. Therefore, we did not smooth the data by five-frame averaging or test the skewness. We have added a detailed description on how the $\Delta F/F$ signals were extracted from the imaging data to the Methods section (lines 720–736).

Reviewer #4: This interesting paper combines two fundamental advances, controlled reaching tasks in marmosets, and two-photon calcium imaging in a primate performing a complex task. As such, the work is of great interest for the field. My only major criticism of the paper is that the analysis of the two-photon data is very limited. In particular, the current analysis does not take advantage of the benefits of two-photon imaging (chronic imaging of the same neurons, and imaging from large groups of neurons with known spatial relationships). The paper could be significantly improved by expanding this part.

Major comments:

- With the exception of a few correlation analyses, there are no statistics for any of the behavioral tasks (are training effects significant, for example?).

We have tested whether the training significantly improved the performance. In the visual-cued pole-pull task, the correlations between the performance (the number of false alarms, hit rate, and number of hits per trial) and the session number were statistically significant ($P < 0.01$ for marmoset A and B, $P < 0.05$ for marmoset C; Fig. 2e–g). In the two-target reaching task, the correlation between the success ratio for reaching target 2 and the session number was statistically significant ($P < 0.01$; Fig. 4b); this was also true for marmoset D, and this information has now been added to the revised manuscript. These statistical results therefore indicate that our training protocols were effective at improving the task performance. We have added these statistical results to the corresponding figure legends.

- The two-photon analysis should be significantly expanded. Some possible venues for additional analyses are described below.

Target reaching task: What are the basic tuning properties of these neurons? Do they respond more for one target direction than the other? Does the tuning change with training? What about activity in error trials?

We defined the movement direction selectivity index (DSI) as the basic tuning index for each neuron in the imaging field, and we have examined its properties. In DSI, a value of 1 indicates that neurons exhibited calcium transients only in movement towards target 1, not in movements towards target 2, and a value of -1 indicates that neurons exhibited calcium transients in only target 2-directed movement. We found that 49 and 22 out of 215 (the cumulative total number) task-relevant neurons from the two marmosets were selective to target 1 and target 2 directed movement respectively

(neurons with DSI > 0.5 and < -0.5 respectively), and that the distribution was slightly biased to positive values (target 1 direction) in both marmosets. DSI values were correlated between days for each marmoset. These results suggest that L2/3 motor cortical neurons with a strong activity preference for the direction of upper-limb movements on a particular day tend to maintain the same preference across days. We have added these results to Fig. 7 in the revised manuscript. The DSI during the error trials was not analyzed, because the cursor trajectory during the trials varied, it was not limited to the direction of the opposite target.

How are responses across neighboring neurons correlated, both during successful and non-successful trials? Do these spatial correlation patterns change with training?

We have calculated the relationship between pairwise correlation coefficients and cellular distance in the imaging fields from the two marmosets. The relationship was negatively correlated, while the activities of neighboring neurons (especially less than 50 μm distance) were highly correlated during successful trials (Supplementary Fig. 7a, b). This negative correlation was conserved across the training sessions (Supplementary Fig. 7c). During the failure trials, this negative correlation was also observed, although statistical significance was not detected in the pooled data and 4 of 6 imaging sessions in one marmoset (Supplementary Fig. 7).

Do certain neurons reliably respond during certain parts of the movement trajectory?

We have calculated the standard deviation (SD) of the trial-to-trial onset timing of the neuronal activity in the two-target reaching task. The activity onset in a trial was calculated as the time when the amplitude of $\Delta F/F$ exceeded 20% of its maximum in that trial. We analyzed the neurons that showed this peak amplitude of > 20% $\Delta F/F$ and an activity onset occurring from 1.5 s before to 1.5 s after the cursor movement onset in more than 2 successful trials. The mean SDs during the movement to target 1 and target 2 were 587.6 ± 26.8 ms and 632.2 ± 27.6 ms respectively ($n = 125$ and 93 neurons from two marmosets, Figure a on the next page). Then, we tested whether the activity onset of neurons with a relatively low SD (< 350 ms) for the onset timing of activity was biased to certain parts of the cursor movement trajectory. As shown in Figure b on the next page, the distribution of the onset timing of these neurons was relatively biased to

the post-movement onset.

Force-field adaptation: Similar to above – what are the basic tuning properties and spatial correlation patterns? Do neurons that change or fail to change their responses in the different blocks cluster?

First, we performed additional imaging experiments by training another marmoset (marmoset A). Then, we analyzed the imaging data from two marmosets. From seven imaging fields in marmoset A, 113 active neurons were extracted and 56 neurons were defined as task-relevant. From eight imaging fields in marmoset D, 425 active neurons were extracted and 163 neurons were defined as task-relevant. The task-relevant neurons were defined as those neurons with significantly higher $\Delta F/F$ signals during the period of -1 to $+2$ s from the movement onset in comparison with the fixation period. Next, we categorized the task-relevant neurons according to their responses during the FF and nFF blocks of the force-field adaptation task (this classification was modified from Li et al., Neuron 30, 593-607, 2001). According to the task-relevant activity averaged over successful trials in each block, we classified these neurons into four

types: “FF-changed”, “BW-changed”, “unchanged” and “other” neurons for each marmoset. FF-changed neurons were defined as task-relevant neurons that exhibited significant differences in the mean task-relevant activity between baseline and FF blocks and between FF and washout blocks, but not between baseline and washout blocks. BW-changed neurons were defined as task-relevant neurons that exhibited significant differences in the mean task-relevant activity between baseline and washout blocks. Unchanged neurons were defined as task-relevant neurons that did not exhibit any significant difference between pairs of the three blocks. Other neurons were defined as the other task-relevant neurons. As the mean cursor trajectory in the FF block deviated substantially from those in the baseline and washout blocks, and the mean cursor trajectory in the baseline block was similar to that in the washout block, we hypothesized that FF-changed neurons might represent the cursor trajectory and BW-changed neurons might be a subset of neurons that changed their activity as the blocks progressed. The FF-changed neurons represented 14.3% and 10.4% of the task-relevant neurons for marmosets A and D respectively. The BW-changed neurons formed 50.0% and 50.9% of the task relevant neurons. These values were significant in comparison with those obtained from shuffled data ($P < 0.05$). We have added these results to Figure 8.

Similar to the analysis in Supplementary Fig. 7, we analyzed the relationship between the pairwise correlation coefficients during successful trials in the three task blocks (baseline, FF and washout) and cellular distance in the imaging fields from the two marmosets. As shown in Figure c below, neighboring neurons at a distance of 50 μm showed highly correlated activities during each task (** $P < 0.01$, Spearman correlation coefficient (CC) between the pairwise CCs and cellular distance). Thus, the strong pairwise correlation between neighboring L2/3 neurons may be conserved in the adaptation task.

We also tested whether BW-changed neurons and either type of FF-changed neurons were spatially clustered, using a nearest neighbor distance (NND) analysis (Diggle P.J., Statistical analysis of spatial point patterns. Oxford Univ. Press, 2003). In this analysis, we first calculated the distances between all pairs of neurons classified as the same type (e.g. FF-changed) in each imaging field. Imaging fields with more than one neuron of the same type ($N > 1$) were analyzed. For each neuron of a given type, the smallest distance to the same type of $N-1$ neurons was assigned to its NND. NND was averaged over the same type of N neurons (mean NND). For each of these neurons, the smallest distance to $N-1$ neurons randomly chosen in the same field was assigned to its shuffled NND, and it was averaged over the same type of N neurons (mean shuffled NND). This shuffling was repeated 1000 times. If the mean NND values were less than the 2.5 percentile of the mean shuffled NNDs, these types of neurons were considered to be clustered. The numbers of fields with a cluster/the number of fields with the same type of multiple neurons ($N > 1$) were: for marmoset A, FF-changed neurons, 0/3, BW-changed neurons, 0/5; for marmoset D, FF-changed neurons, 0/3, BW-changed neurons, 2/8. These results suggest that the neurons that exhibited specific activity changes during the adaptation task might be intermingled, rather than clustered. However, the number of imaging fields with multiple FF-changed neurons was only three for each marmoset, and the number of FF-changed neurons was also low (2, 2, and 3 for marmoset A, and 3, 5, and 6 for marmoset D). We think that many more experiments imaging in three-dimensionally broader areas may be necessary to form firm conclusions on this matter. Therefore, we have not added these results to the revised manuscript.

Dendrites: How do different parts of the same dendritic tree differ in their tuning properties? Are boutons clustered according to their response properties?

In this study, we attempted to image dendrite and axon responses only during the nFF blocks of the adaptation task. In 9 out of 15 imaging fields, we detected movement-related activity of dendrites (dendrite-like structures extracted from the map of correlation coefficients to the cursor trajectory, as shown in Fig. 9c). In addition, in one out of three fields we detected movement-related activity of axons. We have described the experimental result in lines 356–363. To resolve the dendrites and axons, the imaging fields were set to be smaller than those used for the imaging of neuronal somata ($159 \times 159 \mu\text{m}$ for dendrite imaging and $85 \times 85 \mu\text{m}$ for axon imaging). Thus, it was difficult to search long dendritic branches or long axonal arbors with many boutons. To examine how different parts of the same dendritic tree differ in their tuning properties and whether boutons are clustered according to their response properties, we would have to perform many more experiments, and we think that such experiments are beyond the present study. However, we agree that such experiments are important, especially when two-photon imaging is used. We have therefore described the importance of examining the nonlinear dendritic computation in the Discussion section (lines 434 and 435).

Can the authors follow dendrites to their respective somata and compare tuning properties of dendrites with those of somata?

It was difficult to follow the imaged dendrites to their respective neuronal somata, because a constantly bright fluorescent marker was not simultaneously transduced. Therefore, we could not do this in the present study.

Minor comments:

- The introduction is rather unfocused, which makes some of the statements misleading (for example, depending on the brain area neurons can stably represent single parameters; two-photon imaging has been used for a lot more than just population dynamics in working memory etc). In addition, the emphasis on population dynamics in both abstract and introduction seems inappropriate, as these are not further discussed or analyzed in the paper. It would be better to limit the scope of the introduction to movement tasks and the promise of

two-photon imaging in their context.

We are grateful for these comments. In line with them, we have rewritten the introduction to limit its scope and increase its relevancy. In particular, we have removed the description of the population dynamics from both the abstract and introduction.

- It would be helpful to clarify earlier that the initial tasks are not performed with head fixation.

To clarify that some tasks were performed without head fixation, the subtitles of the first and second paragraphs of the Results section are now changed to “Training of marmosets to perform self-initiated upper-limb movement tasks without head fixation” and “Training of a self-initiated pole-pull task in head-fixed marmosets”, respectively. In the revised manuscript, the first paragraph describes the training results without head fixation and the second describes those with head fixation.

- Visually cued pull task:

o Is the threshold across which animals have to pull the pole the same across animals?

During the periods when the parameters were changed day-by-day, we adjusted the threshold depending on the individual marmoset. For the training sessions used for the analysis in Fig. 2, the threshold is fixed to the same values across the animals. To clarify this point, we have added the task parameters during the training sessions to lines 582–584 in the Methods section.

o Do the animals have to hold on to the pole in the blank period?

No, they did not. In the blank period of visual-cued pole-pull task, the animals only needed to keep the pole below the threshold position, and they were allowed to move the pole below the threshold (< 15 mm) without a penalty. When the marmoset released the pole, it returned below the threshold position automatically, because the spring force was applied to the pole during the task. We have described “marmosets needed to keep the pole below the threshold position” in the Methods section (line 572).

o In addition, some discussion of the difference between animals seems warranted.

We have added extra discussion regarding the individual differences in task performance and the duration of the sessions for completing the training of the tasks to the Discussion section (lines 371–379).

- Target reaching task:

o The description of the orthogonality of the two axes in the main text should be improved (it is not immediately clear that leftward is orthogonal to a movement direction toward the animal).

We have clarified this point (lines 152–155).

o Similarly, the main text should explain that the target is straight below the fixation square, otherwise the rest of the analysis is hard to follow.

We have added this point in the main text (lines 150 and 151).

o Why is there a force field applied during the blank period only?

We applied the field during the fixation period as the force would help the marmosets to return the cursor to the center of the fixation square. A force during the reaching period would affect the cursor trajectory and the neuronal activities during the reaching period. Therefore, we applied the force-field only during the fixation period in the reaching tasks.

o The correlation analysis doesn't seem appropriate for these data, which show changes in only very few, early sessions.

We agree with this comment. The task performance rapidly improved within a few early sessions of the training and then appeared to improve only slowly in the following sessions. Therefore, we examined whether the correlation was detectable when the data from sessions 1–3 were removed. We found that the correlation was still significant without these early sessions. CCs between SI and the training session were 0.35 for

marmoset A, $P < 0.05$, and 0.50 for marmoset C, $P < 0.05$. CCs between the mean RMSDs of X and Y coordinates and the session number in marmoset A were -0.50 and -0.46 respectively, $P < 0.01$ for both cases, and CCs in marmoset C were -0.27 , $P = 0.29$ for the X coordinate, and -0.77 , $P < 0.01$ for the Y coordinate. The CC in the mean RMSD of the X coordinate in marmoset C was not significant, but it was also insignificant when all sessions were used. Therefore, the correlation analysis should be appropriate to evaluate the performance improvement during the training of this task.

o There should be some discussion regarding the fact that while the trajectories in both monkeys indeed became straighter, one of the animals failed to perform much above 50% for the majority of the sessions. What happens in these error trials?

As we have indicated, the success rate of marmoset C did not increase above 60% until session 15, although the cursor trajectory did become straighter during the first ten sessions (Fig. 3d). This was because marmoset C had difficulty in stopping and holding the cursor within the target for the set time, and the cursor frequently passed through the target square. Such trials were counted as failures. We have added text describing this to the corresponding figure legend.

- Force-field adaptation:

o The authors state that the force-field remained on for 20 consecutive trials. How many trials total does that constitute? Is there a different failure rate in the early versus late block?

The number of trials included in the FF block of the force-field adaptation task was 27.8 ± 0.7 ($n = 117$ blocks from 58 sessions from three marmosets). The success rates of the task during the early and late FF blocks were $71.0 \pm 2.3\%$ and $80.7 \pm 1.6\%$ respectively (top panel in Fig. 5d). The success rates between the early and late FF blocks were statistically significant ($n = 117$ blocks each, $P < 0.01$, Wilcoxon signed rank test).

Reviewers' comments:

Reviewer #1 (Remarks to the Author):

The authors have adequately addressed my concerns. However, they did not address the issue that the work seems to be more methodological. Methods-type papers are extremely useful so it would be an important contribution to the field.

Reviewer #3 (Remarks to the Author):

The authors addressed all my concerns satisfactorily.

Reviewer #4 (Remarks to the Author):

In this revision, Ebina and colleagues have addressed many of my initial reservations about the paper. A few concerns remain, but require only minor revisions.

- While the authors now give statistics for the behavioral effects, many of them rest on cross-correlation. For many of the tasks, the cross-correlation is able to capture general trends, but matches the actual evolution of learning effects poorly. At the minimum, it would be useful to add other statistics, for example comparisons of the first and last training session only (which would be better in line with the description in the text as well). An alternative would be to fit the data with more appropriate non-linear functions, or to remove the initial sessions from the correlation (as described in the letter to the reviewers). The cross-correlation analysis seems particularly problematic for the force field data, in which one line is fit across data from early and late trials, without regard for the intervening trials.

- The use of an automatic cell-detection algorithm for the 2-photon data is interesting and promising in principle. However, inspection of Figure 7 seems to suggest that the algorithm misses many neurons that are clearly visible in the aggregate image. Since the major emphasis of this paper is the development and test of new tools in the marmoset, it would be useful to provide more data on the quality of the automated algorithm. How many cells does it detect in comparison to manual sorting, for example?

- The criteria used for categorizing cells in the force-field task are problematic because they rely on the failure of certain tests (For example, the failure to find a significant difference between baseline and wash-out does not imply that the response of the neurons are unchanged. Rather, it demonstrates that the null hypothesis of similar responses cannot be rejected). It would be much more convincing to replace the individual tests with one ANOVA across all 3 conditions. Cells that show a significant effect in the ANOVA can then be considered further, with post-hoc tests used to determine the conditions that are different.

We thank all reviewers for their careful consideration of our manuscript and for making helpful comments. Our detailed responses to the reviewer' comments are provided below:

Reviewer #4 (Remarks to the Author):

In this revision, Ebina and colleagues have addressed many of my initial reservations about the paper. A few concerns remain, but require only minor revisions.

- While the authors now give statistics for the behavioral effects, many of them rest on cross-correlation. For many of the tasks, the cross-correlation is able to capture general trends, but matches the actual evolution of learning effects poorly. At the minimum, it would be useful to add other statistics, for example comparisons of the first and last training session only (which would be better in line with the description in the text as well). An alternative would be to fit the data with more appropriate non-linear functions, or to remove the initial sessions from the correlation (as described in the letter to the reviewers). The cross-correlation analysis seems particularly problematic for the force field data, in which one line is fit across data from early and late trials, without regard for the intervening trials.

We have added statistics of the correlation without the initiation session (legends for Figure 2d–g and Figure 3d and 3f). In this statistical test, the changes in the behavioral parameters during training sessions were still significant, except for the correlation between the false alarm number and training session of marmoset C in Fig. 2e. In the two-target reaching task, the values in the initial session in Figure 4b and 4c were not noticeably larger or smaller than those in other sessions and when these values were removed from the correlation calculation, the significance was very similar (success rate for target 1: marmoset A, CC was -0.26 [$P = 0.27$], marmoset D, CC was -0.42 [$P < 0.01$]; success rate for target 2: marmoset A, CC was 0.64 [$P < 0.01$], marmoset D, CC was 0.59 [$P < 0.01$]; trial-to-trial variability for target 1: marmoset A, CCs were -0.22 [$P = 0.36$] for X coordinate and 0.04 [$P = 0.88$] for Y coordinate, marmoset D, CCs were 0.12 [$P = 0.36$] for X coordinate and 0.71 [$P < 0.01$] for Y coordinate; trial-to-trial variability for target 2: marmoset A, CCs were 0.15 [$P = 0.54$] for X coordinate and -0.36 [$P = 0.13$] for Y coordinate, marmoset D, CCs were -0.64 [$P < 0.01$] for X coordinate and -0.36 [$P < 0.01$] for Y coordinate). Therefore, we have not added these results to the revised manuscript.

In the motor adaptation task, the X-axis displacement was compared between the first and last trials in each force-field (FF) block and between the first and last trials in each washout block because the statistical power was sufficient. We found significant differences in both

blocks (FF block, 13.34 ± 0.57 mm vs. 8.40 ± 0.42 mm, $P < 0.01$, Wilcoxon signed-rank test, $n = 117$ blocks in 58 sessions from three marmosets; washout block, -0.97 ± 0.12 mm vs. 0.07 ± 0.05 mm, $P < 0.01$). We have described these results in the Results section (lines 223–225 and 230–232) and removed the linear regression from Figure 5d.

- The use of an automatic cell-detection algorithm for the 2-photon data is interesting and promising in principle. However, inspection of Figure 7 seems to suggest that the algorithm misses many neurons that are clearly visible in the aggregate image. Since the major emphasis of this paper is the development and test of new tools in the marmoset, it would be useful to provide more data on the quality of the automated algorithm. How many cells does it detect in comparison to manual sorting, for example?

ROI contours in Figure 7a correspond to only active neurons with calcium transients. Fluorescent neurons without calcium transients and/or with frequent fluorescent fluctuation were not accepted as active neurons in the CNMF algorithm. Skewness, a distribution's third moment normalized by the cube of the standard deviation, indicates statistical signatures of the distribution of calcium transients in individual neurons (Mukamel et al., *Neuron* 63, 747–760, 2009). This is an easily measurable indicator to pick up active ROIs with a relatively stable baseline and transient positive fluorescence changes at a biologically relevant frequency, making it useful when active neurons are extracted from manually-determined fluorescent ROIs (Mukamel et al., 2009; Bonin et al., *J Neurosci* 31, 18506–18521, 2013; Sadakane et al., *Cell Rep* 13, 1989–1999, 2015). Therefore, in addition to the ROI number, we compared the skewness between manually and automatically detected ROIs. In the first version of our manuscript, 1,516 fluorescent ROIs were manually determined from the total 14 fields from marmoset A (during the two-target reaching task) and marmoset D (during the force-field adaptation task), and out of them, 536 ROIs, which showed >0.55 skewness, were defined as active neurons. The mean skewness of the active neurons was 1.80 ± 0.07 . When using the CNMF algorithm, the search number of ROIs in the field must be set; thus, we set the number of ROIs to 100 ROIs in each field (that is, the total number was 1,400). Then, the CNMF algorithm extracted active neurons from these ROIs according to equipped parameters with default values. 581 active neurons were extracted from the same 14 fields. In these automatically extracted ROIs, the mean skewness was 2.68 ± 0.09 , which was larger than that of ROIs detected manually ($P < 0.01$, Wilcoxon rank-sum test). Thus, in the present version of the manuscript, neurons with higher skewness were analyzed more than those that were manually detected in the first version of our manuscript, although the number of active neurons was similar. The CNMF algorithm has become a popular tool for calcium imaging analyses (e.g.,

Yang et al., eLife 7, e32671, 2018; Klaus et al., Neuron 95, 1171–1180.e7, 2017; Kim et al., Cell Rep 17, 3385–3394, 2016).

In addition, we estimated the skewness of active neurons in our previous study in which cortical neuronal activity was imaged in an anesthetized marmoset (Figure 5 in Sadakane et al., 2015). In that study, 445 fluorescent ROIs were manually determined and 81 ROIs with >0.5 skewness of $\Delta F/F$ were defined as active neurons. The skewness of $\Delta F/F$ in active neurons was 2.23 ± 0.18 ($n = 81$ in three fields from one marmoset). When the CNMF algorithm was applied to those imaging data and the total ROI number for search was set to 450, 87 active ROIs were extracted and the skewness was 2.56 ± 0.17 , which was comparable to that of manually detected active ROIs (skewness, $P = 0.13$, Wilcoxon's rank-sum test). These comparisons suggest that the skewness in the automatically-extracted ROIs was similar to or higher than that in manually detected ones. We have compared the data obtained using the CNMF algorithm in the present study with that in our previous study in which ROIs were manually detected (lines 756–769).

- The criteria used for categorizing cells in the force-field task are problematic because they rely on the failure of certain tests (For example, the failure to find a significant difference between baseline and wash-out does not imply that the response of the neurons are unchanged. Rather, it demonstrates that the null hypothesis of similar responses cannot be rejected). It would be much more convincing to replace the individual tests with one ANOVA across all 3 conditions. Cells that show a significant effect in the ANOVA can then be considered further, with post-hoc tests used to determine the conditions that are different.

We thank you for this comment. We first performed Kruskal-Wallis one-way ANOVA across all three blocks (baseline, FF, and washout blocks). Thirty-eight out of 56 task-relevant neurons in marmoset A and 98 out of 163 task-relevant neurons in marmoset D showed significance. Then, we performed a post-hoc test (Dunn-Sidak test) and classified these neurons into seven groups according to the combination of block pairs with significant differences. Then, the significance of the fraction for each group was tested by comparing it with that obtained from trial shuffled data. The fractions of six groups for marmoset A were significant, as were those of all seven groups for marmoset D. We have replaced Figure 8b with a new figure and rewritten the related results, methods, and legends (lines 334–346, 745–755, 774, and 1179–1188).

REVIEWERS' COMMENTS:

Reviewer #4 (Remarks to the Author):

The authors have addressed my remaining concerns in this revision.